# Active Learning for Derivative-Based Global Sensitivity Analysis with Gaussian Processes

**Syrine Belakaria**
Stanford University

**Benjamin Letham**
Meta

**Janardhan Rao Doppa**
Washington State University

**Barbara Engelhardt**
Stanford University

**Stefano Ermon**
Stanford University

**Eytan Bakshy**
Meta

## Abstract

We consider the problem of active learning for global sensitivity analysis of expensive black-box functions. Our aim is to efficiently learn the importance of different input variables, e.g., in vehicle safety experimentation, we study the impact of the thickness of various components on safety objectives. Since function evaluations are expensive, we use active learning to prioritize experimental resources where they yield the most value. We propose novel active learning acquisition functions that directly target key quantities of derivative-based global sensitivity measures (DGSMs) under Gaussian process surrogate models. We showcase the first application of active learning directly to DGSMs, and develop tractable uncertainty reduction and information gain acquisition functions for these measures. Through comprehensive evaluation on synthetic and real-world problems, our study demonstrates how these active learning acquisition strategies substantially enhance the sample efficiency of DGSM estimation, particularly with limited evaluation budgets. Our work paves the way for more efficient and accurate sensitivity analysis in various scientific and engineering applications.

## 1 Introduction

Sensitivity analysis is the study of how variation and changes in the output of a function can be attributed to distinct sources of variability in the function inputs [13]. More precisely, we seek to determine how changes in each input variable impact the output. Sensitivity analysis can be used for several purposes, including identifying the input variables that are most influential for the function output and those that are least influential [13], and quantifying variable importance in order to explore and interpret a model's behavior [40]. Sensitivity analysis is an important tool in many fields of science and engineering to understand complex, often black-box systems. It has proven particularly important for environmental modeling [31, 41], geosciences [42, 4], chemical engineering [35], biology [15], engineering safety experimentation [29], and other simulation-heavy domains. In these settings, function evaluations often involve time-consuming simulations or costly lab experiments, so it is important to perform the sensitivity analysis with as few function evaluations as possible. For example, sensitivity analysis in environmental modeling can help understand how parameters such as $CO_2$ emissions, temperature increases, and deforestation rates influence the output of climate models. Identifying the variables that most directly impact model outcomes allows researchers to better prioritize efforts in data collection, model refinement, and policy development [31].

Sensitivity analysis can be either local or global. *Local sensitivity analysis* (LSA) [11] studies the effect of perturbations of single input variables at a fixed, nominal point. Input sensitivity is measured only locally to that nominal point and does not take into account any interactions. *Global sensitivity analysis* (GSA) [33], on the other hand, evaluates sensitivity over an entire compact input space

38th Conference on Neural Information Processing Systems (NeurIPS 2024).

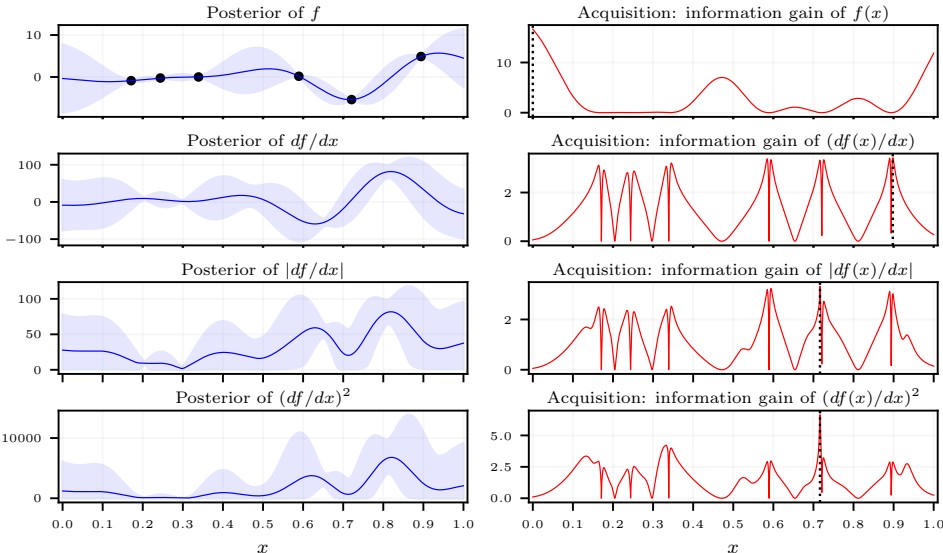

Figure 1: (Left) Posteriors of $f$, $df/dx$, $|df/dx|$, and $(df/dx)^2$ are computed from a GP surrogate given six observations of $f$ (black dots). Posteriors are shown as posterior mean (line) and 95% credible interval (shaded). (Right) Acquisition functions are computed from these posteriors, targeting $f$ and derivative sensitivity measures. Dotted vertical lines show the maximizer. Acquisition functions that directly target DGSMs, not just $f$ generally, are required to learn the DGSMs efficiently.

and includes measures of interactions between the various input dimensions. Here, we focus on GSA. Approaches for GSA can be categorized into two main types: variance-based measures, also referred to as ANOVA decomposition or Sobol methods [28], and derivative-based global sensitivity measures (DGSMs) [17]. Variance-based measures quantify the importance of different input variables based on their contribution to the global variability of the function output. In contrast, DGSMs quantify importance based on the global variability of the function's gradient. They are defined as an integral over the gradients or a function of the gradients across the input space.

DGSMs can often be computed directly from the function. However, for expensive black-box functions, integrating a function of the gradients over the input space is infeasible due to the limited number of function evaluations and a lack of gradient information. In this case, the function is modeled by a surrogate Gaussian process (GP) [6], which allows for tractable computation of both the function surrogate and its gradient. Previous work [6, 17, 21] used random and quasirandom sequences to select the data points for learning the GP; however, these space-filling approaches still require a substantial number of evaluations to accurately estimate the DGSMs.

We show here that DGSMs can be targeted for active learning with information-based acquisition functions that are tractable under GP surrogate models, which are standard for GSA. Applying active learning to DGSM quantities (e.g., gradient, absolute value of the gradient, squared value of the gradient) allows for sensitivity analysis to be performed in a highly sample-efficient manner, suitable for engineering and science applications with small evaluation budgets. To the best of our knowledge, this is the first study proposing active learning acquisition functions directly targeting the DGSM measures.

We illustrate the utility of the proposed acquisition functions for active learning using the classic Forrester [10] test function (Fig. 1). The posteriors of $f$ and $df/dx$ given observations of $f$ are computed from the GP (Sections 2.4 and 2.5). Posteriors of $|df/dx|$ and $(df/dx)^2$ derive from that of $df/dx$ (Section 4.4); the absolute and squared DGSMs are the integrals of these functions (Section 2.3). Acquisition functions give the value of evaluating any particular point under different targets: one quantifies the information gain of the function generally (Section 3), and the others quantify the information gain of the derivative quantities (Section 4.4). The acquisition functions illustrate why active learning strategies that only target learning $f$ are not effective for learning DGSMs. To learn $f$, active learning selects points that are *far from* existing observations, where $f$ is most uncertain.

To learn the derivative—whether raw, absolute, or squared—active learning selects points that are *adjacent to* existing observations, as that adjacency is valuable for derivative estimation.

The contributions of this paper are:

- We introduce the first active learning methods to directly target the quantities used for derivative-based global sensitivity analysis, namely, the gradient, its absolute value, and its square. Our acquisition functions are based on uncertainty reduction and information gain, and we provide tractable approximations of information gain for DGSMs using GP models.

- With a thorough evaluation on synthetic and real-world problems, we show that active learning substantially accelerates GSA in settings with limited evaluation budgets. The implementation for our methods, the baselines, and the synthetic and real-world problems is available in our code (`https://github.com/belakaria/AL-GSA-DGSMs`).

## 2    Background

In this section, we define the problem and provide a review of DGSMs, GPs, and their derivatives. For a thorough review of GSA, see Iooss and Saltelli [13].

### 2.1    Problem Setup

We wish to analyze the sensitivity of a black-box function defined over a compact $d$-dimensional input space $\mathcal{X}$. We suppose that evaluating $f$ at any particular input $\mathbf{x} \in \mathcal{X}$, that is, observing $y = f(\mathbf{x}) + \epsilon$ with $\epsilon$ the observation noise, is expensive. The ground-truth sensitivity measure, which we denote as $S(f, \mathcal{X})$, has a $d$-dimensional output that provides a sensitivity measure for each input dimension $x_i$, $i = 1, \ldots, d$. We estimate $S$ by making $t$ observations of $f$, for which we denote with $X = [\mathbf{x}_1, \ldots, \mathbf{x}_t]$ the set of observed inputs, $Y = [y_1, \ldots, y_t]$ the function evaluations at those inputs, and $\mathcal{D} = \{X, Y\}$ the full observed data. We learn a surrogate model of $f$ from $\mathcal{D}$ and then evaluate $S$ on a surrogate for $f$. Here, we will denote the surrogate as $\hat{f}$; in practice, we will use the GP posterior mean $\hat{f}(\mathbf{x}) = \mu(\mathbf{x})$, which we introduce in Section 2.4. Given the surrogate, we estimate $\hat{S}(f, \mathcal{X}|\mathcal{D}) = S(\hat{f}, \mathcal{X})$.

Our active learning problem is to select the input locations $X$ so that $\hat{S}$ provides the best estimate of the true global measure $S$. We do so sequentially with a budget of $T$ total evaluations. Generally, $\hat{S}$ will better approximate $S$ as the surrogate $\hat{f}$ better approximates $f$. However, when $T$ is small, it is important to consider the particular form of $S$ to design the most effective strategy, as opposed to simply trying to learn a good global surrogate. Here, we develop strategies tailored for $S$ being a DGSM. We introduce DGSMs in Section 2.3, and in Section 4 develop the acquisition strategies.

### 2.2    Bayesian Active Learning

The methods we develop here are cast as Bayesian active learning algorithms. Bayesian active learning is a flexible framework that combines Bayesian inference principles with active learning strategies, where data points are sequentially selected in a sample-efficient manner [12]. The algorithm iterates through three steps. First, at iteration $t$, we build the GP surrogate model from the evaluated data. Second, we apply an acquisition function to the GP posterior to score the utility of unevaluated inputs, and select the input with the highest acquisition value. Third, we evaluate the black-box function on the selected input and augment $\mathcal{D}$ with the new observation. We summarize the process in Algorithm 1. In Section 4, we develop acquisition functions, $\alpha(\cdot)$, that are tailored to learning DGSMs.

### 2.3    Derivative-Based Global Sensitivity Measures

In this section, we provide the background and definitions for our target sensitivity measures, DGSMs. DGSMs are defined as the integral over the input space of a function of the derivative of the black-box function. There are three widely-used gradient functions in DGSMs: the raw gra-

**Algorithm 1** Bayesian Active Learning

**Input:** $\mathcal{X}$, $f(\mathbf{x})$, surrogate model $\mathcal{GP}$, utility function $\alpha(\mathbf{x}, \mathcal{GP})$, total budget $T$.
**Output:** $\mathcal{D}_T, \mathcal{GP}$.

1: Initialize data $\mathcal{D}_0$ with $T_0$ starting observations.
2: **for** each iteration $t \in [T_0, T]$ **do**
3:     Fit the surrogate model $\mathcal{GP}(\mathcal{D}_{t-1})$ using $\mathcal{D}_{t-1}$ .
4:     Select the next input for evaluation by maximizing the acquisition function,
    $\mathbf{x}^* \leftarrow \arg\max_{\mathbf{x} \in \mathcal{X}} \alpha(\mathbf{x}, \mathcal{GP}(\mathcal{D}_{t-1}))$.
5:     Evaluate the black-box function to observe $y^* = f(\mathbf{x}^*)$.
6:     Update data $\mathcal{D}_t = \mathcal{D}_{t-1} \cup \{(\mathbf{x}^*, y^*)\}$.
7: **end for**

dient, the absolute value of the gradient, and the square of the gradient [18]:

$$S_R(f, \mathcal{X})_i = \frac{1}{|\mathcal{X}|} \int_{\mathcal{X}} \left( \frac{\partial f(\mathbf{x})}{\partial x_i} \right) d\mathbf{x}, \quad S_{Ab}(f, \mathcal{X})_i = \frac{1}{|\mathcal{X}|} \int_{\mathcal{X}} \left| \frac{\partial f(\mathbf{x})}{\partial x_i} \right| d\mathbf{x},$$

$$S_{Sq}(f, \mathcal{X})_i = \frac{1}{|\mathcal{X}|} \int_{\mathcal{X}} \left( \frac{\partial f(\mathbf{x})}{\partial x_i} \right)^2 d\mathbf{x}.$$

In the remainder of the paper, we will refer to these quantities as the raw DGSM, absolute DGSM, and squared DGSM. These quantities may also be defined with non-uniform densities on $\mathcal{X}$.

For the purpose of evaluating input sensitivity, the raw DGSM is considered uninformative due to a phenomenon known as *the cancellation effect*. In nonmonotonic functions, positive parts of the gradient cancel out negative parts when integrated over the entire input space, leading to a small value for the raw DGSM even for important dimensions. The most commonly used DGSMs in practice are the absolute and squared DGSMs, which avoid the cancellation effect. The squared DGSM is especially popular because of its connection to the variance of the gradient [17]. Computing DGSMs requires computing a $d$-dimensional integral over $\mathcal{X}$. This integration is usually done via Monte Carlo (MC) or quasi-Monte Carlo (QMC) sampling [7].

### 2.4 GP Surrogates for Sensitivity Analysis

When function evaluations are expensive, the integrals over the input space required to compute the DGSMs may not be evaluated tractably from $f$. Moreover, if $f$ is black-box, we may not have access to its gradients. Both of these issues can be avoided by using a surrogate function for $f$.

GSA of expensive, black-box functions is usually done using a GP surrogate model [6]. GPs are characterized by a mean function $m : \mathcal{X} \to \mathbb{R}$ and a kernel function covariance $\mathcal{K} : \mathcal{X} \times \mathcal{X} \to \mathbb{R}$. A GP prior for the function, $f \sim \mathcal{GP}(m, \mathcal{K})$, means that the function values at any finite set of inputs are jointly normally distributed. For any input $\mathbf{x}_* \in \mathcal{X}$, the function value at that input has a normally-distributed posterior $f(\mathbf{x}_*)|\mathcal{D} \sim \mathcal{N}(\mu_*, \sigma_*^2)$, whose predictive mean and variance are:

$$\mu_* = \mathcal{K}_{\mathbf{x}_*, X} K_{\mathcal{D}}^{-1} (Y - m_X) + m_{\mathbf{x}_*}, \quad \sigma_*^2 = \mathcal{K}_{\mathbf{x}_*, \mathbf{x}_*} - \mathcal{K}_{\mathbf{x}_*, X} K_{\mathcal{D}}^{-1} \mathcal{K}_{X, \mathbf{x}_*},$$

where $\mathcal{K}_{\mathbf{x}_*, \mathbf{x}_*} = \mathcal{K}(\mathbf{x}_*, \mathbf{x}_*)$, $\mathcal{K}_{\mathbf{x}_*, X} = [\mathcal{K}(\mathbf{x}_*, \mathbf{x}_j)]_{j=1}^t$, $m_{\mathbf{x}} = m(\mathbf{x})$, and $K_{\mathcal{D}} = \mathcal{K}_{X, X} + \eta^2 I$, with $\mathcal{K}_{X, X} = \mathcal{K}(X, X)$ and $\eta^2$ the observation noise variance of $y$ [30]. We introduce the short-hand notation $\mu_* := \mu(\mathbf{x}_*)$ and $\sigma_*^2 := \sigma(\mathbf{x}_*)^2$ for the posterior mean and variance functions.

GPs are differentiable when using any twice-differentiable kernel function $\mathcal{K}$ and a differentiable mean function $m$. The gradient of the GP provides a tractable estimate of the gradient of the expensive black-box function $f$ under commonly-used kernels such as the ARD RBF. DGSMs can then be computed in a fast and scalable way on the posterior of $f$ [6].

### 2.5 Derivatives of Gaussian Processes

GPs are closed under linear operations, therefore the derivative of a GP is itself a GP [30]. Since $f$ is defined over a $d$-dimensional input space, the model's gradient has a $d$-dimensional output. Under a GP prior, the joint distribution between (potentially noisy) observations of $f$ and the gradient of $f$

at a new point $\mathbf{x}_*$ is as follows [30]:

$$\begin{bmatrix} Y \\ \nabla f(\mathbf{x}_*) \end{bmatrix} \sim \mathcal{N} \left( \begin{bmatrix} m_X \\ \nabla m_{\mathbf{x}_*} \end{bmatrix}, \begin{bmatrix} K_{\mathcal{D}} & \nabla_{\mathbf{x}_*} \mathcal{K}_{X,\mathbf{x}_*} \\ \nabla_{\mathbf{x}_*} \mathcal{K}_{\mathbf{x}_*,X} & \nabla^2_{\mathbf{x}_*} \mathcal{K}_{\mathbf{x}_*,\mathbf{x}_*} \end{bmatrix} \right).$$

Given the observed data $\mathcal{D}$ as before, the gradient at $\mathbf{x}_*$ has a multivariate normal distribution: $\nabla f(\mathbf{x}_*)|\mathcal{D} \sim \mathcal{N}(\mu'_*, \Sigma'_*)$, where

$$\mu'_* = \nabla m_{\mathbf{x}_*} + \nabla_{\mathbf{x}_*} \mathcal{K}_{\mathbf{x}_*,X} K_{\mathcal{D}}^{-1}(Y - m_X), \tag{1}$$

$$\Sigma'_* = \nabla^2_{\mathbf{x}_*} \mathcal{K}_{\mathbf{x}_*,\mathbf{x}_*} - \nabla_{\mathbf{x}_*} \mathcal{K}_{\mathbf{x}_*,X} K_{\mathcal{D}}^{-1} \nabla_{\mathbf{x}_*} \mathcal{K}_{X,\mathbf{x}_*}. \tag{2}$$

Here, $\mu'_* = \mu'(\mathbf{x}_*)$ and $\Sigma'_* = \Sigma'(\mathbf{x}_*)$ are shorthand for the posterior mean and covariance functions of the gradient. Note that the posterior for the derivative may be obtained from observations only of $f$, and does not require direct observations of the derivative. The greatest computational expense in computing the GP posterior is the matrix inversion in $K_{\mathcal{D}}^{-1}$, which has complexity $\mathcal{O}(t^3)$. This same term is also the most expensive term in the posterior for the derivative. Consequently, once the posterior of the GP has been computed, the computation of the derivative does not increase the overall complexity. Given the posterior of the derivative, the DGSMs are estimated by substituting the gradient of the function with the predictive mean of the gradient from (1).

## 3    Related Work

The GP surrogate allows for computing DGSMs with a limited set of function evaluations. However, in budget-restricted experiments, the GP will only provide a faithful representation of the sensitivity of $f$ if the right set of inputs are evaluated. There has been limited work on efficiently selecting the inputs that lead to accurate DGSM estimation, particularly with a limited evaluation budget.

**Random and space-filling designs:** The most common approach for estimating DGSMs with a GP is to evaluate the function on either a random set of inputs or with a space-filling design. For the latter, quasirandom sequences like scrambled Sobol sequences [26] and Latin hypercube sampling [24] are two common choices. Space-filling designs are effective for GSA with a sufficiently large evaluation budget, however, as we will see below, they fail when the budget is limited.

**General uncertainty reduction methods:** Several Bayesian active learning approaches have been developed for the purpose of reducing global uncertainty about $f$. Information-based strategies that select the point that produces the largest information gain about a function's outputs are popular and effective for global identification of $f$ [36, 16, 12]. Other global active learning approaches are based on variance reduction [34] and expected improvement (EI) [19]. These general-purpose active learning strategies have been applied to the GSA problem. Pfingsten [27] used global predictive variance reduction as the active learning target for the purpose of GSA; Chauhan et al. [3] applied the EI criterion to GSA. These approaches are designed to generally minimize uncertainty of $f$, and do not specifically target improvement of any particular GSA measure. Acquisition functions that target $f$ are not sufficient to learn the DGSMs efficiently on a budget (Figure 1).

**Active learning involving derivatives:** Some work has incorporated derivatives into active learning for problems unrelated to GSA. Salem et al. [32] and Spagnol et al. [37] use sensitivity measures to eliminate variables during Bayesian optimization. Erickson et al. [9] and Marmin et al. [23] include a derivative term in an acquisition function for learning non-stationary functions. Wycoff et al. [43] do active subspace identification with an acquisition function targeting the outer product of the gradient. See the Appendix for further discussion and an empirical comparison to these methods.

**Active learning for GSA:** There has been limited work on applying active learning to GSA measures. Existing work considers only the Sobol index (variance-based measures). Le Gratiet et al. [20] applied variance reduction directly to the Sobol index. However, this could not be done in closed form and required expensive simulations within each active learning step. More recently, Chauhan et al. [3] developed an analytic improvement criterion that targets the numerator of the Sobol index. In this work, we propose the first active learning acquisition functions directly targeting DGSMs.

# 4 Bayesian Active Learning for Derivative-Based Sensitivity Analysis

In this section, we develop active learning acquisition functions that target the DGSM measures. We derive acquisition functions following three general strategies. The *maximum variance* acquisition functions select the point with the largest posterior variance in the quantity of interest, indicating the point with the most uncertainty. The *variance reduction* acquisition functions measure how much an observation at a point will reduce the variance at that point in expectation over the possible outcomes of the observation. Finally, the *information gain* acquisition functions quantify the expected reduction in entropy of the posterior of the quantity of interest for each point. The latter two strategies require computing the look-ahead distribution for the derivative, which we introduce in Section 4.1. Finally, we discuss global look-ahead acquisition functions that measure the impact of evaluating an input on the posterior across the entire input space.

## 4.1 The Derivative Look-Ahead Distribution

Effective active learning often relies on computing *look-ahead* distributions that predict the impact that making a particular observation will have on the model. For our purposes, we wish to predict the impact that observing $f$ at a candidate point $\mathbf{x}_*$ will have on the model's derivatives at that location. Conditioned on the observations $\mathcal{D}$, $f(\mathbf{x}_*)$ and $\frac{\partial f(\mathbf{x}_*)}{\partial x_i}$ have a bivariate normal joint distribution for each input dimension $i$. The well-known formula for bivariate normal conditioning provides the look-ahead distribution [22]: $\frac{\partial f(\mathbf{x}_*)}{\partial x_i} \mid f(\mathbf{x}_*) = y_*, \mathcal{D} \sim \mathcal{N}\left(\mu_{*,i}'^{\ell}, (\sigma_{*,i}'^{\ell})^2\right)$, where the look-ahead mean and variance are

$$\mu_{*,i}'^{\ell} = \mu_{*,i}' + \frac{\tilde{\sigma}_{*,i}}{\sigma_*^2}(y_* - \mu_*), \quad (\sigma_{*,i}'^{\ell})^2 = (\sigma_{*,i}')^2 - \left(\frac{\tilde{\sigma}_{*,i}}{\sigma_*}\right)^2, \tag{3}$$

with $\tilde{\sigma}_{*,i} = \mathrm{Cov}[f(\mathbf{x}_*), \frac{\partial f(\mathbf{x}_*)}{\partial x_i}|\mathcal{D}]$ the posterior covariance between $f$ and the derivative at $\mathbf{x}_*$, and $(\sigma_{*,i}')^2 = \sigma_i'(\mathbf{x}_*)^2 = [\Sigma'(\mathbf{x}_*)]_{ii}$ the posterior variance of the derivative. As before, we use the notational short-hand $\sigma_{*,i}'^{\ell} = \sigma_i'^{\ell}(\mathbf{x}_*)$ and $\mu_{*,i}'^{\ell} = \mu_i'^{\ell}(\mathbf{x}_*)$. This result holds when $y_*$ is a noisy observation by replacing $\sigma_*^2$ with $\eta^2 + \sigma_*^2$. Remarkably, the look-ahead variance is independent of the actual observed $y_*$, so acquisition functions that are based on the future variance of the derivative can be computed exactly in closed form. In the Appendix, we provide the look-ahead posterior distribution of the derivative of $f$ at any point in the input space after observing $f$ at $\mathbf{x}_*$.

## 4.2 Gradient Acquisition Functions

**Maximum variance.** The posterior variance of each derivative is given in (2). The maximum derivative variance acquisition function uses the sum of the variances across dimensions to find points with high total uncertainty in the derivatives: $\alpha_{\mathrm{DV}}(\mathbf{x}) = \sum_{i=1}^{d} \sigma_i'(\mathbf{x})^2$.

**Variance reduction.** The derivative variance reduction acquisition computes the expected reduction in variance of the derivatives produced by making an observation of $f$ at $\mathbf{x}$:

$$\alpha_{\mathrm{DV_r}}(\mathbf{x}) = \sum_{i=1}^{d} \sigma_i'(\mathbf{x})^2 - \mathbb{E}_y[\sigma_i'^{\ell}(\mathbf{x})^2] = \sum_{i=1}^{d} \sigma_i'(\mathbf{x})^2 - \sigma_i'^{\ell}(\mathbf{x})^2,$$

where $\sigma_i'^{\ell}(\mathbf{x})^2$ is the look-ahead variance of the derivative, from (3). The expectation is dropped because the look-ahead variance is independent of the observed $y$ at the candidate point.

**Information gain.** We express our derivative information gain acquisition function as the sum of information gains for each derivative. Let $H_i'(\mathbf{x}) = h\left(\frac{\partial f(\mathbf{x})}{\partial x_i}|\mathcal{D}\right)$ be the differential entropy of each derivative posterior and $H_i'^{\ell}(\mathbf{x}) = h\left(\frac{\partial f(\mathbf{x})}{\partial x_i}|\mathcal{D}, f(\mathbf{x}) = y\right)$ the look-ahead entropy. The Gaussian entropy is well-known [5], and since it is independent of the mean, the look-ahead entropy is independent of $y$. The information gain, in nats, is:

$$\alpha_{\mathrm{DIG}}(\mathbf{x}) = \sum_{i=1}^{d} H_i'(\mathbf{x}) - \mathbb{E}_y\left[H_i'^{\ell}(\mathbf{x})\right] = \sum_{i=1}^{d} \log\left(\sigma_i'(\mathbf{x})\right) - \log\left(\sigma_i'^{\ell}(\mathbf{x})\right).$$

### 4.3 Absolute Gradient Acquisition Functions

**Maximum variance.** The absolute value of a normal distribution is the *folded normal distribution* [38], whose mean and variance, $\mu'_{i_{Ab}}(\mathbf{x})$ and $\sigma'_{i_{Ab}}(\mathbf{x})^2$, are analytical and can be computed from the moments of the corresponding normal distribution. Using those results, the posterior of $\left|\frac{\partial f(\mathbf{x})}{\partial x_i}\right|$ has mean and variance:

$$\mu'_{i_{Ab}}(\mathbf{x}) = \sqrt{\frac{2}{\pi}}\sigma'_i(\mathbf{x})e^{-\frac{1}{2}r_i^2(\mathbf{x})} + \mu'_i(\mathbf{x})\left(1 - 2\Phi(-r_i(\mathbf{x}))\right),$$

$$\sigma'_{i_{Ab}}(\mathbf{x})^2 = \mu'_i(\mathbf{x})^2 + \sigma'_i(\mathbf{x})^2 - \mu'_{i_{Ab}}(\mathbf{x})^2,$$

where $\Phi$ is the standard normal CDF and we have denoted $r_i(\mathbf{x}) = \frac{\mu'_i(\mathbf{x})}{\sigma'_i(\mathbf{x})}$. We define the maximum variance acquisition for the absolute value of the derivative as: $\alpha_{\text{DAbV}}(\mathbf{x}) = \sum_{i=1}^d \sigma'_{i_{Ab}}(\mathbf{x})^2$.

**Variance reduction.** The look-ahead variance for the absolute value of the derivative, denoted $\sigma'^\ell_{i_{Ab}}(\mathbf{x})^2$, can be computed by plugging the look-ahead moments from (3) into the formula for the folded normal variance. However, unlike for the raw derivative, this variance depends on $\mu'^\ell_i(\mathbf{x})$ and is thus a function of $y$, making the expectation in the variance reduction formula intractable. We follow the strategy of Lyu et al. [22] and approximate $\mathbb{E}_y[\sigma'^\ell_{i_{Ab}}(\mathbf{x})^2]$ with a plug-in estimator, fixing $y = \mu(\mathbf{x})$. Plugging this estimator into (3) gives an estimate for the look-ahead derivative mean that is independent of $y$, denoted $\hat{\sigma}'^\ell_{i_{Ab}}(\mathbf{x})^2$, and it follows that

$$\alpha_{\text{DAbV}_r}(\mathbf{x}) = \sum_{i=1}^d \sigma'_{i_{Ab}}(\mathbf{x})^2 - \mathbb{E}_y[\sigma'^\ell_{i_{Ab}}(\mathbf{x})^2] \approx \sum_{i=1}^d \sigma'_{i_{Ab}}(\mathbf{x})^2 - \hat{\sigma}'^\ell_{i_{Ab}}(\mathbf{x})^2.$$

**Information gain.** The differential entropy of the folded normal distribution is not available in closed form. Tsagris et al. [38] provide an approximation using a truncated Taylor series. However, we show in the Appendix that it is numerically poorly behaved in this application. We introduce a novel approximation for the folded normal differential entropy, in nats:

$$H_i^{ab}(\mathbf{x}) = H'_i(\mathbf{x}) - \frac{(\mu'_{i_{Ab}}(\mathbf{x})^2 - \mu'_i(\mathbf{x})^2)\pi\log(2)}{2\sigma'_i(\mathbf{x})^2}.$$

The derivation of this approximation and an evaluation of accuracy alongside other approximations are in the Appendix. The look-ahead entropy $H_i^{ab,\ell}$ is estimated using the same plug-in strategy as for variance reduction, to remove dependence on $y$ and get

$$\alpha_{\text{DAbIG}}(\mathbf{x}) = \sum_{i=1}^d H_i^{ab}(\mathbf{x}) - H_i^{ab,\ell}(\mathbf{x}). \tag{4}$$

### 4.4 Squared Gradient Acquisition Functions

**Maximum variance.** Let $Q = \left(\frac{\partial f(\mathbf{x})}{\partial x_i}\right)^2$ and $Z = \frac{Q}{\sigma'_i(\mathbf{x})^2}$. The posterior of $Z$ has a non-central $\chi^2$ distribution $Z|\mathcal{D} \sim \chi'^2_1\left(\frac{\mu'_i(\mathbf{x})^2}{\sigma'_i(\mathbf{x})^2}\right)$. This allows computing the posterior variance of the squared derivative using the known moments of the noncentral $\chi^2$ distribution: $\sigma'_{i_{sq}}(\mathbf{x})^2 = 4\sigma'_i(\mathbf{x})^2\mu'_i(\mathbf{x})^2 + 2\sigma'_i(\mathbf{x})^4$. As before, we construct the maximum variance acquisition function as $\alpha_{\text{DSqV}}(\mathbf{x}) = \sum_{i=1}^d \sigma'_{i_{sq}}(\mathbf{x})^2$.

**Variance reduction.** As with the absolute value, the look-ahead variance for the square of the derivative depends on the observed value $y$ via the term $\mu'^\ell_i(\mathbf{x})$, making the expectation in variance reduction intractable. We again approximate the variance reduction with a plug-in estimator in (3), substituting $\mu'_i(\mathbf{x})$ for $\mu'^\ell_i(\mathbf{x})$ to estimate a look-ahead variance $\hat{\sigma}'^\ell_{i_{sq}}$ independent of $y$. The variance reduction can then be computed as:

$$\alpha_{\text{DSqV}_r}(\mathbf{x}) = \sum_{i=1}^d \sigma'_{i_{sq}}(\mathbf{x})^2 - \mathbb{E}_y[\sigma'^\ell_{i_{sq}}(\mathbf{x})^2] \approx \sum_{i=1}^d \sigma'_{i_{sq}}(\mathbf{x})^2 - \hat{\sigma}'^\ell_{i_{sq}}(\mathbf{x})^2.$$

**Information gain.** Using properties of entropy [5], the differential entropy of the squared derivative follows from that of the noncentral $\chi^2$ distribution as:

$$H_i^{sq}(\mathbf{x}) = h\left(Q|\mathcal{D}\right) = h\left(Z|\mathcal{D}\right) + 2\log(\sigma_i'(\mathbf{x})). \tag{5}$$

In the Appendix we develop two approximations for the entropy of the noncentral $\chi^2$. As shown there, the most effective approach derives from an earlier approximation of the quantile function for the noncentral $\chi^2$ by Abdel-Aty [1]. Our novel entropy approximation relies on recent analytical expressions for the expected logarithm by Moser [25], as detailed in the Appendix. We plug the approximated noncentral $\chi^2$ entropy into (5) to obtain an entropy for the squared normal, and then the information gain acquisition, $\alpha_{\mathrm{DSqIG}}(\mathbf{x})$, follows analogously to $\alpha_{\mathrm{DAbIG}}$.

$$\alpha_{\mathrm{DSqIG}}(\mathbf{x}) = \sum_{i=1}^{d} H_i^{sq}(\mathbf{x}) - H_i^{sq,\ell}(\mathbf{x}). \tag{6}$$

### 4.5 Global Variance Reduction and Information Gain

The acquisition functions described so far all evaluate the impact of an observation at $\mathbf{x}_*$ only on the posterior at $\mathbf{x}_*$. We also considered global look-ahead acquisitions, which evaluate the impact of an observation at $\mathbf{x}_*$ on the posterior across the entire input space. These acquisition functions are expressed as an integral of the acquisition functions already described. We provide their full expressions, their evaluation, and a discussion about their complexity and performance in the Appendix.

## 5 Experiments

### 5.1 Experimental Setup

We compared the proposed active learning acquisition functions for DGSMs to space-filling and general uncertainty reduction approaches. We refer to quasirandom sequences as QR, variance maximization of $f$ as fVAR, and information gain about $f$ [i.e., BALD, 12] as fIG. Since our experiments use noiseless observations of $f$, variance reduction of $f$, fV$_r$, is equal to fVAR. For the acquisition functions developed here, we use the following acronyms (Section 4): max variance of the raw, absolute, and squared derivatives are DV, DAbV, and DSqV; corresponding variance reduction acquisitions are DV$_r$, DAbV$_r$, and DSqV$_r$; and information gain acquisitions are DIG, DAbIG, and DSqIG. For the absolute and squared derivative information gains, we evaluated two different entropy approximations, labeled as e.g. DSqIG$_1$ and DSqIG$_2$, with the corresponding descriptions in the Appendix.

We used synthetic and real-world problems for evaluation. For synthetic experiments we used a family of functions designed specifically for evaluating sensitivity analysis measures [18, 17]: Ishigami1 ($d = 3$), Ishigami2 ($d = 3$), Gsobol6 ($d = 6$), a-function ($d = 6$), Gsobol10 ($d = 10$), Gsobol15 ($d = 15$) and Morris ($d = 20$). Ground-truth DGSMs are available for these problems. We additionally used other general-purpose synthetic functions where sensitivity might be challenging to estimate [8]: Branin ($d = 2$), Hartmann3 ($d = 3$) and Hartmann4 ($d = 4$). For these functions, we numerically estimated ground-truth DGSMs. The results for Hartmann3, Gsobol6, and a-function are given in the Appendix. We considered three real-world design problems. The *Car Side Impact Weight* problem simulates the impact of $d = 7$ design variables on the weight of a car, to study the impact of weight on accident scenarios. Design variables are the thickness of pillars, the floor, cross members, etc. We also used the *Vehicle Safety* problem, which has two functions: the weight and the acceleration of the vehicle. Both are functions of $d = 5$ design variables describing the thickness of frontal reinforcement materials. We study the two functions as independent problems. Ground truth DGSMs for these problems were estimated numerically.

We study settings with limited evaluation budgets. Quasirandom sequences are known to perform well given enough data [6, 3]. Here, we focus on the restrictive case where we initialize our experiments using five random inputs and run 30 iterations of active learning. Our results are averaged over 50 replicates from different initial points, and we report the mean and two standard errors over replicates. Our primary evaluation metric is root mean squared error (RMSE) of the DGSM estimate versus ground truth. All acquisition functions were implemented in BoTorch [2] and were designed to be auto-differentiable and efficiently optimized with gradient optimization.

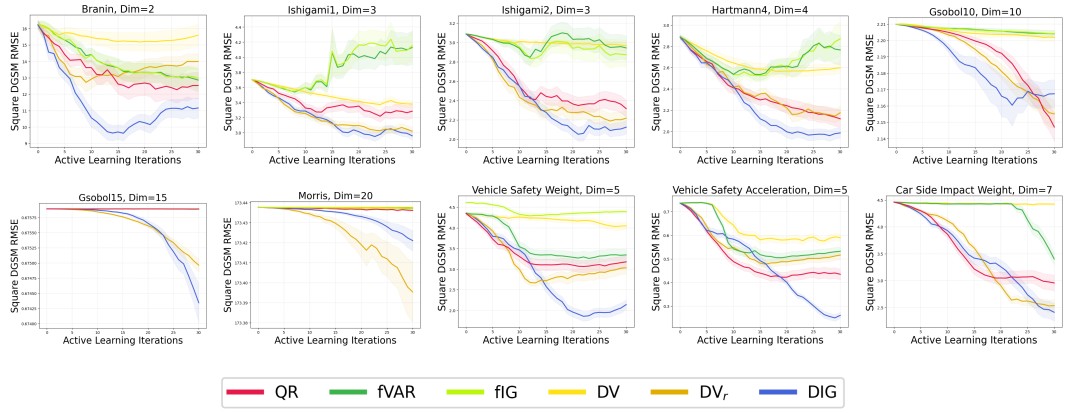

Figure 2: RMSE (mean over 50 runs, two standard errors shaded) for learning the DGSM, for 10 test problems. Results are shown for active learning methods targeting the raw derivative. Active learning targeting the derivative consistently outperformed space-filling designs and active learning targeting $f$. Derivative information gain was generally the best-performing acquisition function.

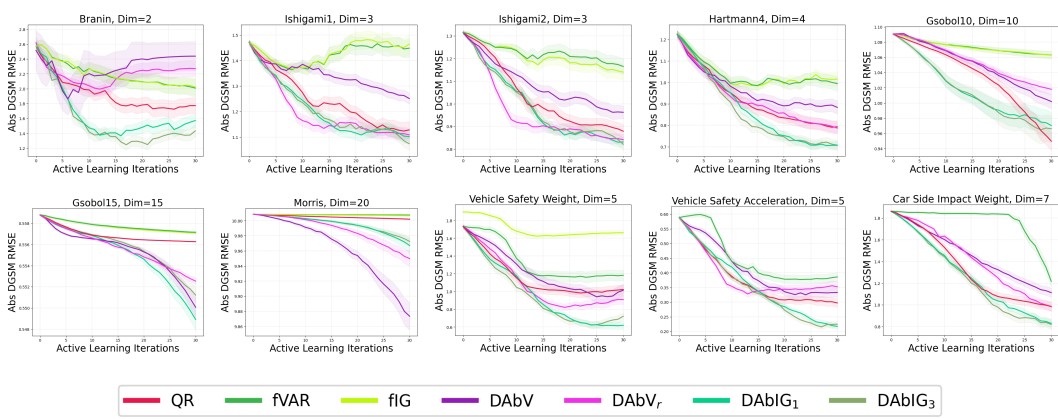

Figure 3: RMSE results as in Fig. 2, here for the absolute derivative acquisition functions. These also outperformed the baselines, with absolute derivative information gain generally the best.

## 5.2 Results and Discussion

Experimental results for the new acquisition functions are separated by their targets for clarity: Fig. 2 for the raw derivative, Fig. 3 for the absolute derivative, and Fig. 4 for the squared derivative. Across this wide set of problems, the active learning targeting DGSM quantities developed here consistently outperformed quasirandom sequences (QR) and active learning methods that target learning about $f$ (fVAR and fIG). The DGSM information gain acquisition functions (DIG, $\text{DAbIG}_1$, $\text{DSqIG}_1$) performed best in the majority of experiments.

There were rare instances in which RMSE increased with active learning iteration. The explore/exploit trade-off is fundamental to active learning. During exploration, adding a data point in one part of the space may cause a global adjustment in the model predictions that can cause errors in another area. With more exploration and data, the model self-corrects and RMSE will decrease.

In the high-dimensional problems (GSobol15 and Morris), we see a substantial degradation of performance for the baseline methods (QR, fVAR, and fIG), with little reduction of RMSE across iterations. Active learning targeting the DGSM quantities, on the other hand, continued to perform well in high dimensions. On these problems the derivative max variance acquisition functions (DV, DAbV, and DSqV) were competitive with or outperformed the derivative information gain acquisition functions. We hypothesize this is due to the myopic nature of the one-step look-ahead used for information gain.

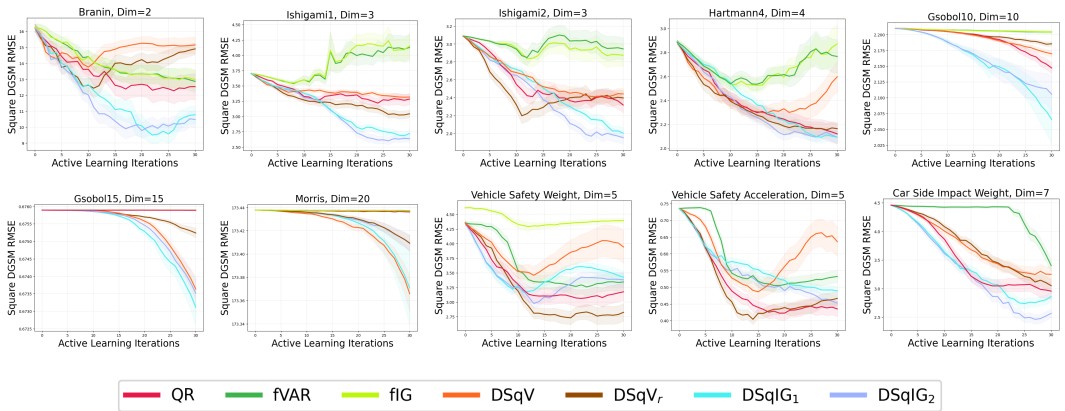

Figure 4: RMSE results as in Fig. 2, here for the squared derivative acquisition functions. As with the other derivative active learning approaches, these outperformed the baselines, and squared derivative information gain generally performed best.

Information gain is maximized near existing points (Fig. 1), and so in high dimensions, one-step look-ahead is not sufficiently exploratory to capture the whole function. Max variance is naturally more exploratory and thus can perform better in high-dimensional settings. However, the derivative information gain acquisitions still outperformed the baselines on these problems. In practice, derivative information gain measures are likely to be the best choice, possibly ensembled with a derivative max variance acquisition in high dimensional problems [39].

### 5.3 Additional Results

In the Appendix, we also evaluate the ability of active learning to identify the correct ordering of variable importance, by using normalized discounted cumulative gain (NDCG) [14] as the evaluation metric. The results were generally consistent with what is seen with RMSE. We further evaluated global versions of these same acquisition functions, and found that they did not substantially improve over the results shown here, while creating a large computational burden. Finally, we evaluated a heuristic baseline that incorporates the insight of Fig. 1 by using a space-filling design plus small perturbations of those same points. This did not significantly improve performance over QR.

## 6 Conclusions

In this work, we developed a collection of active learning methods that directly target DGSMs. These strategies substantially enhanced the sample efficiency of DGSM estimation when compared to quasirandom search and even compared to active learning strategies targeting $f$. Information gain about the derivative (raw, absolute, or squared) was generally the best approach.

Our work paves the way for additional work on active learning for DGSMs in several directions. Although both variance reduction and information gain approaches perform well in high dimensions, information gain approaches might benefit from increased exploration by using a two-step or batch look-ahead to select pairs of points in acquisition function optimization. Our acquisition functions also all take the form of a sum over dimensions. Computing the entropy of multivariate posteriors for the gradient is another possible avenue. Active learning for DGSMs could also be developed for non-Gaussian models. Finally, DGSMs have been linked via several lower/upper bound inequalities to ANOVA-based sensitivity indices [17]. Understanding the impact of active learning for DGSMs on ANOVA-based sensitivity indices is another useful direction for future work.

**Acknowledgments.** Belakaria is supported by a Stanford Data Science Postdoctoral Fellowship. Doppa is supported in part by USDA-NIFA funded AgAID Institute award #2021-67021-35344 and NSF CAREER award IIS-1845922.

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

# Active Learning for Derivative-Based Global Sensitivity Analysis with Gaussian Processes (Supplementary Material)

## A  Additional Discussion of Related Work

Here we give further discussion of lines of work that incorporate derivatives into active learning, albeit not for the purpose of GSA.

**Goal-Oriented Sequential Design.** Salem et al. [32] proposed an optimization approach where at each step the dimensionality is reduced by identifying unimportant features. Unimportant variables are fixed while simultaneously optimizing the important ones with expected improvement. These variables are identified through their lengthscales. As part of theoretical analysis, the paper proves an asymptotic relationship between lengthscale and the square DGSM: as lengthscale goes to infinity, DGSM for that parameter goes to zero. This work does not propose an active learning approach for DGSMs but rather uses DGSMs as a metric to asses the quality of the dimensionality reduction approach. Spagnol et al. [37] used sensitivity analysis to eliminate variables during optimization. The sensitivity analysis is goal-oriented rather than global, by applying the Hilbert-Schmidt independence criterion to portions of the function below a particular output quantile. Sensitivity measures are part of the algorithm and are assumed to be accurate, the paper does not study learning them.

**Active learning in the presence of non-stationarity.** Erickson et al. [9] developed a method for active learning of a non-stationary function that specifically targets areas of high gradient, although without trying to estimate the gradient globally. Variance reduction of the function serves as the acquisition, but weighted by the estimated derivative and its variance. The method thus focuses on minimizing the variance at points with large derivatives, but does not target learning the derivative itself. Marmin et al. [23] proposed an active learning approach for non-stationary functions using warping. The acquisition strategy uses both variance reduction and derivatives, following a similar idea as Erickson et al. [9] that areas of high gradient will be helpful for learning the function.

**Active Subspaces Learning.** Wycoff et al. [43] proposed a set of three methods (`Trace`, `Var1`, and `Var2`) for active subspace identification. While the goal is different from our proposed methods, the acquisition functions developed in the paper do aim to learn a quantity of the derivatives, as in our work. Specifically, the acquisitions attempt to learn the expectation of the outer product of the gradient. The elements of the diagonal of the outer product of the gradient are the squared DGSMs, so learning the outer product will learn a function of the squared DGSMs, albeit without targeting them specifically.

Because these methods target learning a function of the gradient, we compared against them on three representative problems. We used the reference implementation of the methods, from the R package `activegp`. The results are shown in Fig. 5. For clarity and due to the large number of methods, this figure includes only QR and DIG alongside `Trace`, `Var1`, and `Var2`; performance relative to other methods can be seen by comparing the results to Figs. 2, 3, and 4. The figure shows that active subspace learning methods do not perform as well as derivative information gain. While they do target a related quantity, learning the active subspace is not the most effective approach to learning the squared DGSM.

## B  Entropy Approximations

Here we develop approximations for the differential entropies of the absolute and squared derivative posteriors.

### B.1  Folded Normal Entropy

The posterior of the absolute value of the derivative in this setting is a folded normal distribution. Specifically, we consider $X \sim \mathcal{N}(\mu, \sigma^2)$ and are then interested in the entropy of $Y = |X|$. We describe three approximations for the entropy of a folded normal.

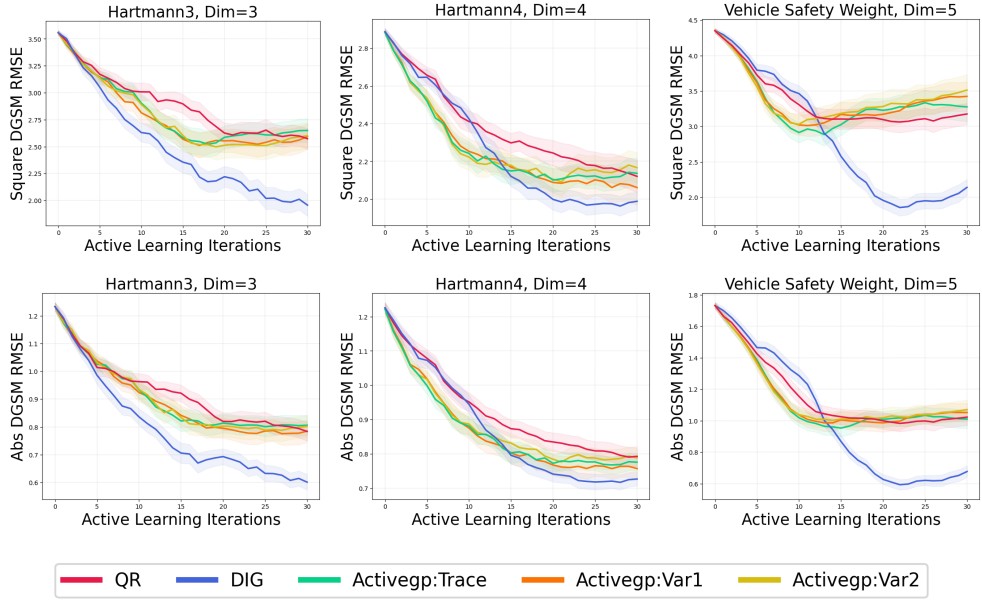

Figure 5: An empirical evaluation of active subspace methods on the task of learning the square and absolute DGSM, with derivative information gain, derivative square information gain, and quasirandom as points of comparison.

### B.1.1 Folded Normal: Approximation 1

The folded normal distribution has two limiting forms. For $\mu$ different than 0, as the variance of the (pre-folded) normal $\sigma^2$ goes to zero, the amount of probability density being folded goes to zero, and thus the distribution approaches a normal distribution with that same variance. The limit of the entropy in this case is the normal distribution entropy (here in nats):

$$h(X) = \frac{1}{2} + \frac{1}{2} \log(2\pi\sigma^2).$$

On the other hand, as the variance becomes large relative to the mean, the proportion of density being folded goes to 1/2, and the limiting distribution is the half-normal distribution. For $Z$ with a half-normal distribution, the differential entropy is analytic: $h(Z) = h(X) - \log(2)$. Furthermore, the variance of $Y$ is upper-bounded by the variance of the $X$, and lower-bounded by the variance of $Z$.

We approximate the folded normal entropy as a convex combination of these two limiting distributions, using the variance bounds. Specifically, we take $h(Y) \approx h(X) - w \log(2)$, where $w$ goes to 0 as the distribution approaches a normal, and $w$ goes to 1 as the distribution approaches a half-normal. This is achieved by taking:

$$
\begin{aligned}
w &= \frac{\text{Var}(Y) - \text{Var}(Z)}{\text{Var}(X) - \text{Var}(Z)} \\
&= \frac{\mu^2 + \sigma^2 - \mu_Y^2 - \sigma^2(1 - 2/\pi)}{\sigma^2 - \sigma^2(1 - 2/\pi)} \\
&= \frac{\pi(\mu_Y^2 - \mu^2)}{2\sigma^2},
\end{aligned}
$$

where, of course, $\mu_Y = \mathbb{E}[Y]$, which is known analytically. The final result for the entropy approximation is thus, in nats,

$$h(Y) \approx \frac{1}{2}\left(1 + \log(2\pi\sigma^2) - \frac{(\mu_Y^2 - \mu^2)}{\sigma^2}\pi\log(2)\right).$$

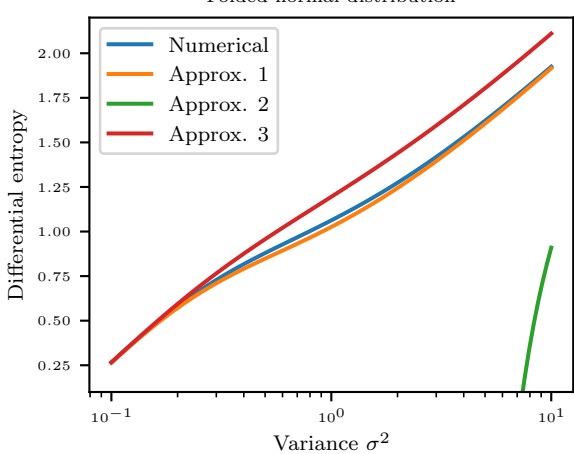

Figure 6: Differential entropy of a folded normal distribution as a function of the underlying normal variance $\sigma^2$. The figure compares a high-precision numerical estimate with three approximation schemes, of which scheme 1 provides a high-fidelity analytic approximation.

### B.1.2 Folded Normal: Approximation 2

To our knowledge, the only analytic approximation already found in the literature is that of Tsagris et al. [38, Equation 40], which uses a truncated Taylor series:

$$h(Y) \approx \log(\sqrt{2\pi\sigma^2}) + \frac{1}{2} + \frac{\mu^2 - \mu\mu_Y}{\sigma^2}$$
$$- \sum_{n=1}^{\infty} \frac{(-1)^{n+1}}{n} \exp\left(\frac{(\mu - 2n\mu)^2 - \mu^2}{2\sigma^2}\right) \left(2 - \Phi\left(-\frac{\mu}{\sigma} - \frac{2n\mu}{\sigma^3}\right) - \Phi\left(\frac{\mu}{\sigma} - \frac{2n\mu}{\sigma^3}\right)\right).$$

Tsagris et al. [38] recommend truncating the infinite series after 3 terms, which is what we did here.

### B.1.3 Folded Normal: Approximation 3

For a given variance, differential entropy is maximized by the normal distribution. A (first and second) moment-matched normal distribution thus provides an upper-bound for differential entropy that can be used as an approximation for any distribution. In the case of the folded normal, this produces the approximation

$$h(Y) \approx \frac{1}{2} + \frac{1}{2} \log(2\pi\sigma_Y^2)$$
$$= \frac{1}{2} + \frac{1}{2} \log(2\pi) + \log\left(\mu^2 + \sigma^2 + \mu_Y^2\right).$$

### B.1.4 Evaluation of Folded Normal Entropy Approximations

Fig. 6 shows an evaluation of the accuracy of the three approximations described for the differential entropy of the folded normal distribution. They are compared to a high-precision numerical estimation of the differential entropy obtained via numerical integration. Entropy values are shown as a function of the variance $\sigma^2$, for fixed $\mu = 1$.

The numerically estimated entropy in Fig. 6 clearly shows the limiting behavior described in Sec. B.1.1: for small $\sigma^2$ it converges to the normal upper bound, and for large $\sigma^2$ it converges to one bit of entropy less than the normal upper bound (the half-normal lower bound). Approximation 1 (convex combination of normal and half-normal entropies) provides a close estimate of the true entropy. Approximation 2 (truncated Taylor series) is numerically unstable for this range of variances and so provided a poor approximation and was unsuitable for optimization in active learning.

We included both approximations 1 and 3 in the experiments in the main text (DAbIG$_1$ and DAbIG$_3$ respectively), and they produced similar active learning performance.

## B.2 Squared Normal Entropy

The posterior of the square of the derivative in this setting is the square of a normal distribution. For $X \sim \mathcal{N}(\mu, \sigma^2)$, we are now interested in the entropy of $Y = X^2$. As described in the main text, if we let $Z = \frac{Y}{\sigma^2}$, then $Z$ has a noncentral $\chi^2$ distribution $Z \sim \chi_1'^2(\lambda)$, with noncentrality parameter $\lambda = \frac{\mu^2}{\sigma^2}$. Because

$$h(Y) = h(Z) + \log(\sigma^2), \tag{7}$$

the problem of computing the entropy of the squared normal reduces to that of computing the entropy of the noncentral $\chi^2$ distribution, with one degree of freedom. We describe two approximations for this entropy.

### B.2.1 Noncentral Chi-Squared: Approximation 1

Abdel-Aty [1, 'first approx.'] showed that the following provides an accurate approximation of the quantiles of a noncentral $\chi^2$ with one degree of freedom:

$$\left( \frac{Z}{1+\lambda} \right)^{\frac{1}{3}} \sim \mathcal{N}(1 - C, C),$$

where $C = \frac{2}{9} \frac{1+2\lambda}{(1+\lambda)^2}$. Let $g(Z) = \left( \frac{Z}{1+\lambda} \right)^{\frac{1}{3}}$. Because $g$ is a bijection,

$$h(g(Z)) = h(Z) + \mathbb{E}\left[ \log \left| \frac{\partial g}{\partial Z} \right| \right]. \tag{8}$$

We know $h(g(Z))$ because $g(Z)$ has a normal distribution. For the expectation term,

$$\mathbb{E}\left[ \log \left| \frac{\partial g}{\partial Z} \right| \right] = \mathbb{E}\left[ \log \left| \frac{1}{3} \left( \frac{Z}{1+\lambda} \right)^{-\frac{2}{3}} \left( \frac{1}{1+\lambda} \right) \right| \right]$$

$$= \mathbb{E}\left[ -\log(3) - \frac{2}{3}\log(Z) - \frac{1}{3}\log(1+\lambda) \right]$$

$$= -\log(3) - \frac{1}{3}\log(1+\lambda) - \frac{2}{3}\mathbb{E}[\log(Z)].$$

In a recent work, Moser [25, Equation 54] provide an analytic result for the log expectation of a noncentral $\chi^2$ random variable:

$$\mathbb{E}[\log(Z)] = {}_2F_2\left( 1, 1, \frac{3}{2}, 2, -\frac{\lambda}{2} \right)\lambda - \gamma - \log(2),$$

where $\gamma \approx 0.577$ is Euler's constant and ${}_2F_2$ is a generalized hypergeometric function. Combining these results yields an approximation for the noncentral $\chi^2$ differential entropy:

$$h(Z) \approx \frac{1}{2} + \gamma + \log(2) + \frac{1}{2}\log\left( \frac{4\pi(1+2\lambda)}{9(1+\lambda)^2} \right) - {}_2F_2\left( 1, 1, \frac{3}{2}, 2, -\frac{\lambda}{2} \right)\lambda.$$

### B.2.2 Noncentral Chi-Squared: Approximation 2

As in Sec. B.1.3, we can approximate the entropy of the noncentral $\chi^2$ using the moment-matched normal distribution upper bound to its entropy. The variance of a noncentral $\chi^2$ distribution with one degree of freedom is simply $2(1+2\lambda)$, and so the upper bound on the entropy is:

$$h(Z) \approx \frac{1}{2} + \frac{1}{2}\log(4\pi(1+2\lambda)).$$

### B.2.3 Evaluation of Noncentral Chi-Squared Entropy Approximations

Figure 7 shows an evaluation of the accuracy of the two approximations described for the differential entropy of the noncentral $\chi^2$ distribution with one degree of freedom. They are compared to a high-precision numerical estimation obtained via numerical integration. The left panel shows the

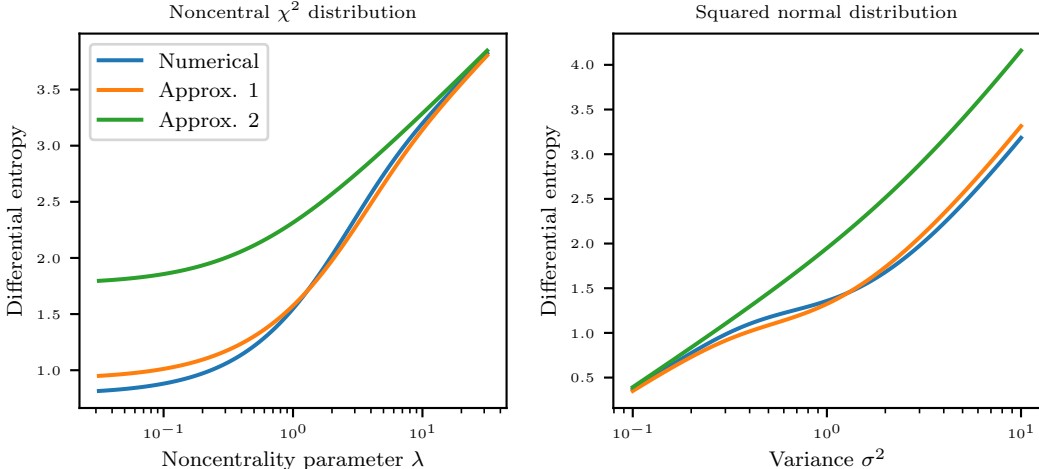

Figure 7: Differential entropy of (left) a noncentral $\chi^2$ distribution as a function of its parameter $\lambda$, and (right) a squared normal distribution as a function of the underlying normal variance $\sigma^2$. The figure compares a high-precision numerical estimate with two approximation schemes, of which scheme 1 provides a high-fidelity analytic approximation.

noncentral $\chi^2$ entropy as a function of the noncentrality parameter $\lambda$. The right panel shows the entropy of the squared normal distribution computed via (7), as a function of $\sigma^2$ and with fixed $\mu = 1$. Approximation 1 (using the quantile function of Abdel-Aty) provides a close estimate of the true entropy, for both the noncentral $\chi^2$ distribution and after transformation to the squared normal entropy. The normal upper bound in approximation 2 is accurate only for small variances, when the amount of reflection at the origin is small.

Both of these approximations were also included in the experiments in the main text, as DSqIG$_1$ and DSqIG$_2$. They usually performed similarly, though DSqIG$_2$ performed slightly better in some problems despite being a worse approximation for the actual underlying entropy. Approximation 2 overestimates the entropy, and thus the information gain, at inputs with high predictive variance. This will generally make the acquisition more exploratory, which seems to have been beneficial on those problems.

## C   Computational Complexity

All of our acquisition functions have closed-form expressions. Computing the GP posterior is $\mathcal{O}(t^3 + t^2d)$, with $t$ the number of data and $d$ the dimensionality of the function. Here the $t^3$ comes from inverting $k_X$, and the $t^2d$ comes from multiplying it with the gradient of $k_{x*,X}$. Given that, the remaining computation will all scale linearly with $d$, since it is just applying various formulae to the posteriors for each dimension.

## D   Global Look-Ahead Acquisition Functions

Our look-ahead acquisition functions were all local, in that they evaluated the impact of an observation at $\mathbf{x}_*$ only on the posterior at $\mathbf{x}_*$. In this section, we consider the global look-ahead acquisitions where we evaluate the impact on the posterior across the entire input space. These acquisition functions are expressed as integral over the previously proposed local acquisition functions. Their complexity is $\mathcal{O}(t^3 + t^2dM)$, with $M$ defined as the granularity of the integration. We provide the results and discussion in Appendix E.

### D.1   Derivative Look-Ahead Distribution at Different Input Locations

We wish to predict the impact that observing $f$ at a candidate location $\mathbf{x}_*$ will have on the model's derivatives at a different location $\mathbf{x}_+$. This will allow us to select a point $\mathbf{x}_*$ that most improves the

model of the derivatives in the overall search space, in expectation. Under a GP, this look-ahead distribution is tractable. Conditioned on the observations $\mathcal{D}$, $f(\mathbf{x}_*)$ and $\frac{\partial f(\mathbf{x}_+)}{\partial x_i}$ have a bivariate normal joint distribution for each input dimension $i$. The formula for bivariate normal conditioning once again provides the look-ahead distribution:

$$\frac{\partial f(\mathbf{x}_+)}{\partial x_i} \bigg| f(\mathbf{x}_*) = y_*, \mathcal{D} \sim \mathcal{N}\left(\mu'^\ell_{+|*,i}, (\sigma'^\ell_{+|*,i})^2\right),$$

where the look-ahead mean and variance are respectively

$$\mu'^\ell_{+|*,i} = \mu'_{+,i} + \frac{\tilde{\sigma}_{+*,i}}{\sigma_*^2}(y_* - \mu_*), \quad (\sigma'^\ell_{+|*,i})^2 = (\sigma'_{+,i})^2 - \left(\frac{\tilde{\sigma}_{+*,i}}{\sigma_*}\right)^2 \tag{9}$$

with $\tilde{\sigma}_{+*,i} = \mathrm{Cov}[f(\mathbf{x}_*), \frac{\partial f(\mathbf{x}_+)}{\partial x_i}|\mathcal{D}]$ the posterior covariance between $f$ at $\mathbf{x}_*$ and the derivative at $\mathbf{x}_+$, and $(\sigma'_{+,i})^2 = \sigma'_i(\mathbf{x}_+)^2 = [\Sigma'(\mathbf{x}_+)]_{ii}$ the posterior variance of the derivative. We use a similar notation short-hand $\sigma'^\ell_{+|*,i} = \sigma'^{\ell^*}_i(\mathbf{x}_+)$ and $\mu'^\ell_{+|*,i} = \mu'^{\ell^*}_i(\mathbf{x}_+)$ as the look-ahead posterior mean and variance at the input $\mathbf{x}_+$ if $(\mathbf{x}_*, y_*)$ were added to the data.

## D.2 Global Look-Ahead Acquisition Functions

Global look-ahead acquisition functions are constructed by using the distribution in (9) as the look-ahead distribution when constructing the acquisition functions as described in Section 4. The acquisition then provides the value that making an observation at $\mathbf{x}_*$ will have for the model predictions at $\mathbf{x}_+$. To make the acquisition function global, we integrate over the input space. For example, global derivative variance reduction is:

$$\alpha_{\text{IDV}_r}(\mathbf{x}_*) = \frac{1}{|\mathcal{X}|} \int_{\mathcal{X}} \sum_{i=1}^d \sigma'_i(\mathbf{x})^2 - \mathbb{E}_{y_*}[\sigma'^{\ell^*}_i(\mathbf{x})^2] d\mathbf{x} = \frac{1}{|\mathcal{X}|} \int_{\mathcal{X}} \sum_{i=1}^d \sigma'_i(\mathbf{x})^2 - \sigma'^{\ell^*}_i(\mathbf{x})^2 d\mathbf{x}.$$

For all of these global acquisition functions, the integral over the input space is estimated with QMC sampling, meaning the impact of the observation at $\mathbf{x}_*$ is evaluated on a global reference set of size $M$, over which the acquisition function is averaged.

# E  Additional Results and Discussion

## E.1  Additional Synthetic Problems

Figs. 8, 9, and 10 provide experimental results for the Hartmann3, Gsobol6, and a-function problems. The results are largely consistent with those of the other problems in the main text, the main exception being the stronger than usual performance of QR on the Gsobol6 problem. The derivative max variance methods also perform better than usual on this problem, suggesting that a higher degree of exploration is important.

The main text evaluated performance of the raw and squared derivative acquisition functions on the squared DGSM, and the absolute derivative acquisition functions on the absolute DGSM. Fig. 11 shows the performance of all acquisitions on both DGSMs, for all 13 problems. On most problems, the squared and absolute derivative information gains performed similarly, without clear distinction between learning the absolute or squared DGSM.

## E.2  Running Times

Table 1 gives the wall time running times for acquisition optimization. Generally information gain methods are more expensive, with the squared derivative information gain particularly expensive due to the use of a hypergeometric function in its approximation. However, even the maximum time for that method (214 seconds) is fast relative to the time-consuming function evaluation setting that is the focus of this paper. It is interesting to also note that running time generally increases with dimension but not always. This is because the optimization time will depend on the shape of the acquisition function surface and how hard the optimization is, which is separate from the dimensionality of the problem.

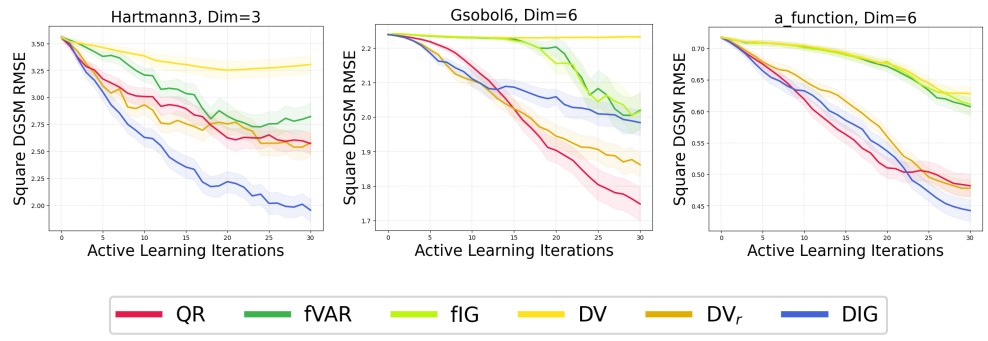

Figure 8: Additional experiment problems for the results in Fig. 2.

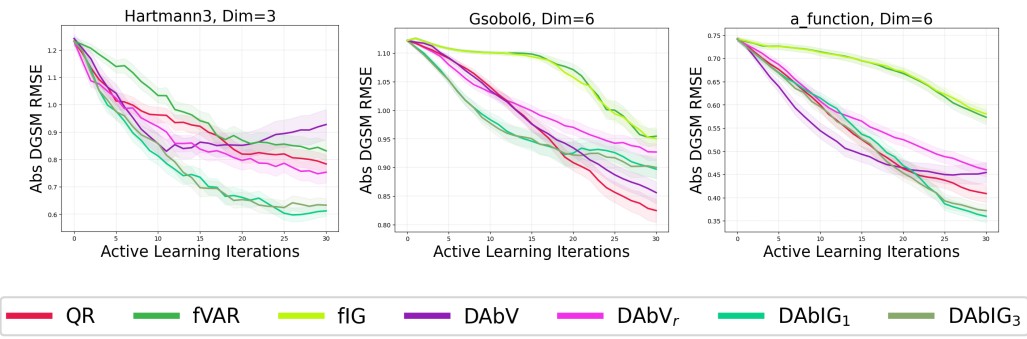

Figure 9: Additional experiment problems for the results in Fig. 3.

### E.3 Results and Discussion for Ranking Metrics

In Figure 12, we show the performance evaluated with NDCG. NDCG accounts for the position of each dimension, giving higher importance to dimensions at the top of the list, and is normalized to a scale from 0 to 1, where 1 represents perfect recovery of the order. RMSE is generally a better metric since it shows the ability of the method to converge to the true values of DGSMs, and because it is the most commonly used metric in the literature. As far as we are aware, NDCG has not been used before as a performance metric. However, in some applications, the ranking of the variables is important for practitioners, for instance when doing parameter selection for downstream tasks. We thus included a ranking metric to provide another perspective into the results. When evaluated with this ranking metric, we find that active learning is able to efficiently discover the correct ordering of variables according to their importance, and that derivative information gain acquisitions continue to perform generally best.

Fig. 13 shows the results of the ranking metric NDCG for two of the synthetic functions, a-function and Morris. For these problems, there is a clear instability in their ranking results. These two particular problems have several dimensions with the very similar DGSMs values that thus cannot be reliably sorted. Thus the ranking of the variables changes frequently over iterations leading to the instability in the curves. This is an issue of NDCG as a metric, and shows that care should be taken when using it for this purpose.

### E.4 Results and Discussion for Global Look-ahead Acquisition Functions

In this study, we compare the local look-ahead acquisition functions proposed in the main text to the global look-ahead acquisition functions described in Appendix D. We denote the global variance reduction of the absolute and squared derivatives as $\mathrm{IDAbV_r}$ and $\mathrm{IDSqV_r}$, and global information gain of the raw and squared derivatives as $\mathrm{IDIG}$ and $\mathrm{IDSqIG}$.

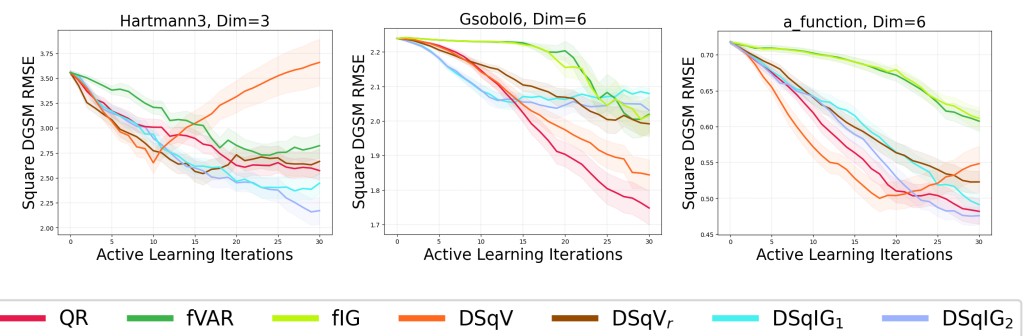

Figure 10: Additional experiment problems for the results in Fig. 4.

| Method | Ishigami1 | Gsobol6 | Gsobol10 | Gsobol15 | Morris |
|---|---|---|---|---|---|
| fVAR | 0.44±0.03 | 0.39±0.02 | 0.19±0.01 | 0.21±0.02 | 0.2±0.02 |
| fIG | 0.43±0.03 | 0.37±0.01 | 0.29±0.04 | 0.17±0.02 | 0.16±0.01 |
| DV | 0.62±0.08 | 0.31±0.01 | 0.45±0.07 | 0.34±0.04 | 0.16±0.01 |
| DV$_r$ | 1.43±0.2 | 3.43±0.49 | 10.56±1.78 | 6.32±1.06 | 3.54±0.51 |
| DIG | 8.59±1.15 | 16.75±0.46 | 24.6±2.5 | 28.15±4.75 | 13.92±3.78 |
| DAbV | 1.37±0.13 | 4.26±0.3 | 3.93±0.39 | 5.31±0.63 | 5.87±0.7 |
| DAbV$_r$ | 2.4±0.3 | 9.69±1.19 | 13.31±1.16 | 8.48±1.43 | 3.96±0.42 |
| DSqV | 1.58±0.22 | 4.23±0.62 | 12.18±4.23 | 16.1±2.03 | 6.95±1.0 |
| DSqV$_r$ | 2.84±0.64 | 6.66±0.88 | 8.8±1.77 | 6.29±0.9 | 9.0±2.29 |
| DSqIG | 96.3±19.52 | 214.39±30.84 | 180.5±24.79 | 122.27±18.61 | 95.24±28.67 |

Table 1: Average acquisition function optimization times across the active learning loop for each method and for a representative sample of problems.

As discussed above, the complexity of the global acquisition function depends on the granularity of the numerical integration. Given that the granularity of the integration $M$ needs to be high for numerical accuracy, the computation time and memory requirement becomes prohibitive, especially in high dimensions. Figs. 14 and 15 show that despite the extra computational cost, the global acquisition functions have comparable performance for variance reduction and lower performance for information gain than their local versions. Given the high computational cost incurred by the integration, we found the local acquisition functions to be more practical.

### E.5 Adding Perturbations to the Space-Filling Design

Fig. 1 showed that derivative active learning methods selected points adjacent to existing points, which aids in estimating the derivative. A reasonable heuristic for adapting a space-filling design to derivative estimation is to sample pairs of points, in which one point in each pair is a small perturbation of the other. We implemented that heuristic to see if it could improve performance of the baseline.

We generated a quasi-random sequence and then interleaved it with points that were small perturbations of the point before it. Specifically, for all n odd, we took $\mathbf{x}^n$ from the quasi-random sequence, and then took $\mathbf{x}^{n+1} = \mathbf{x}^n + \epsilon$, where $\epsilon \sim \mathcal{N}(\mathbf{0}, I\eta)$, with $\eta = 0.01$ and box bounds scaled to $[0, 1]$. That is, each dimension was given a small Gaussian perturbation on the value from the previous iteration. We denote the method as QR-P. Fig. 16 shows the performance results alongside QR and some active learning methods. This approach did significantly improve performance on one of the problems, but generally the results were comparable to QR without the interleaved pairing. This approach does rely on the perturbation hyperparameter $\eta$, and the "right" choice likely depends on the problem.

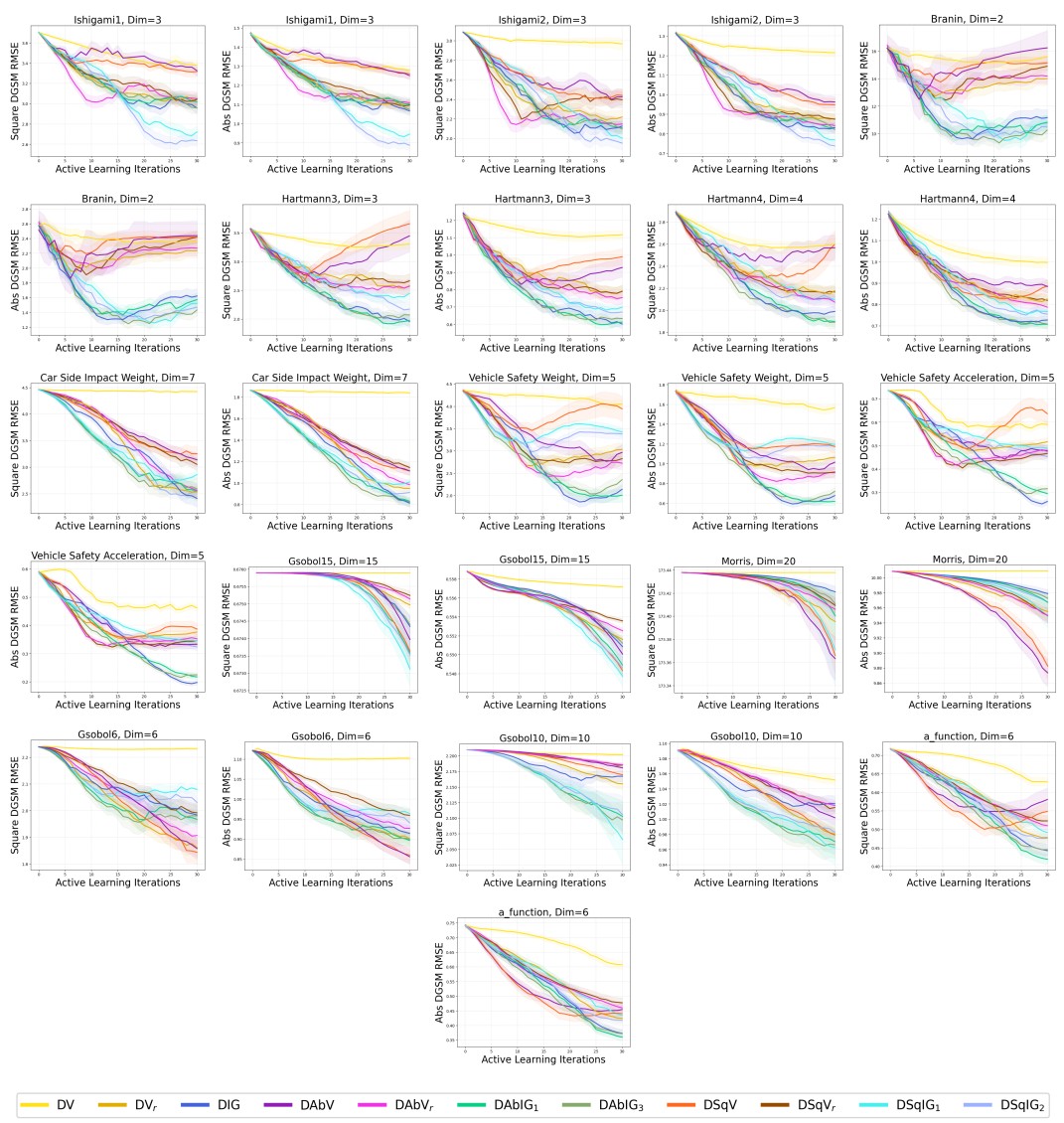

Figure 11: All acquisition functions evaluated on all benchmark problems for both the absolute and squared DGSMs. A superset of the results in Figs. 2, 3, and 4.

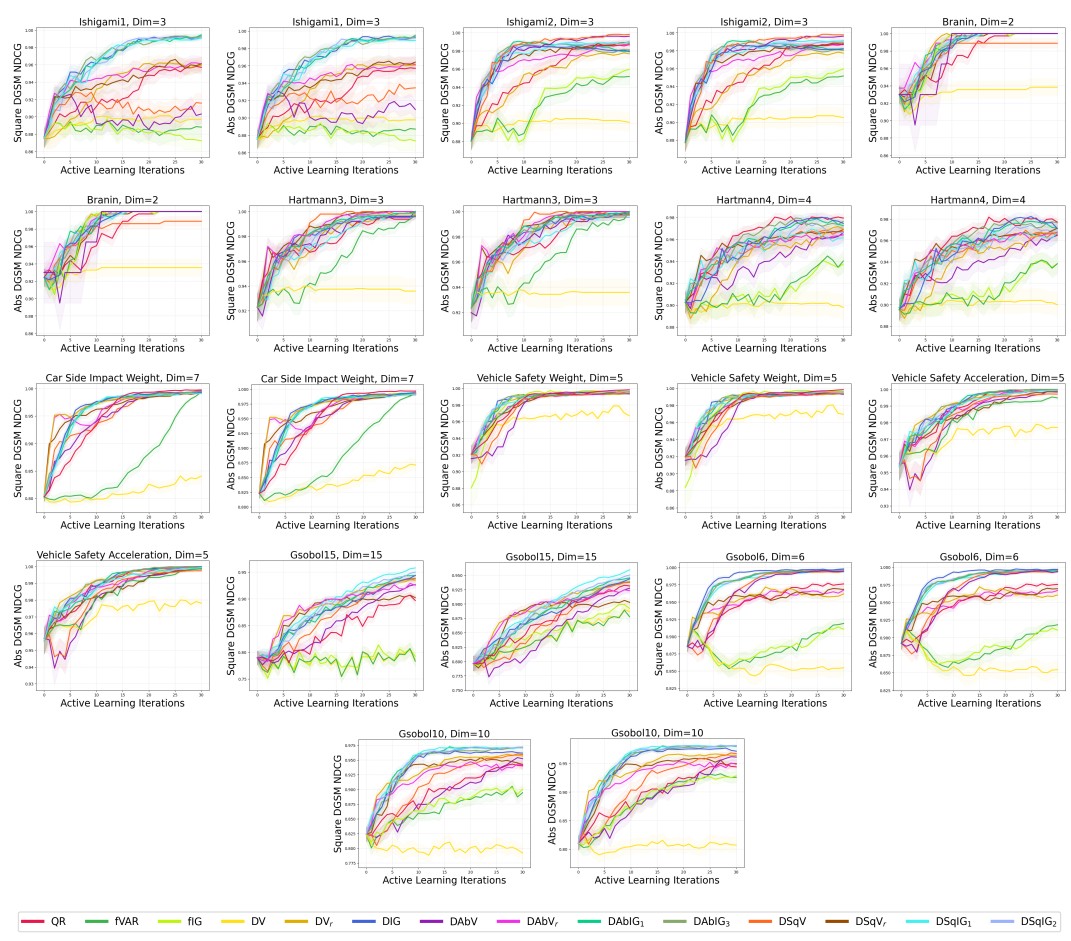

Figure 12: Synthetic and real-world experiments evaluated using NDCG. The NDCG of the square DGSMs and absolute DGSMs is reported with the mean and two standard errors of 50 repeats.

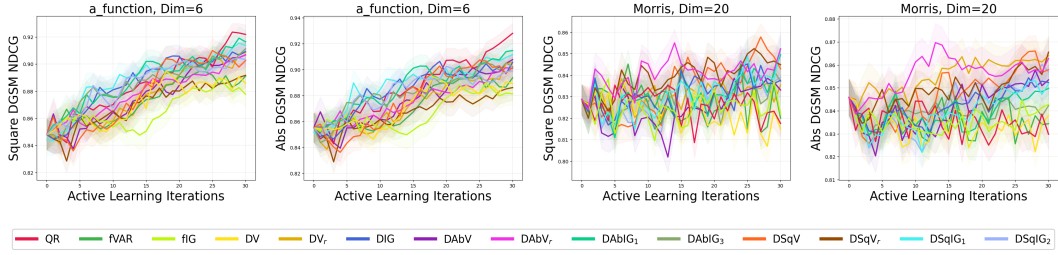

Figure 13: Synthetic experiments with multiple variables holding the same ranking evaluated using NDCG. The NDCG of the square DGSMs and absolute DGSMs is reported with the mean and two standard errors of 50 repeats.

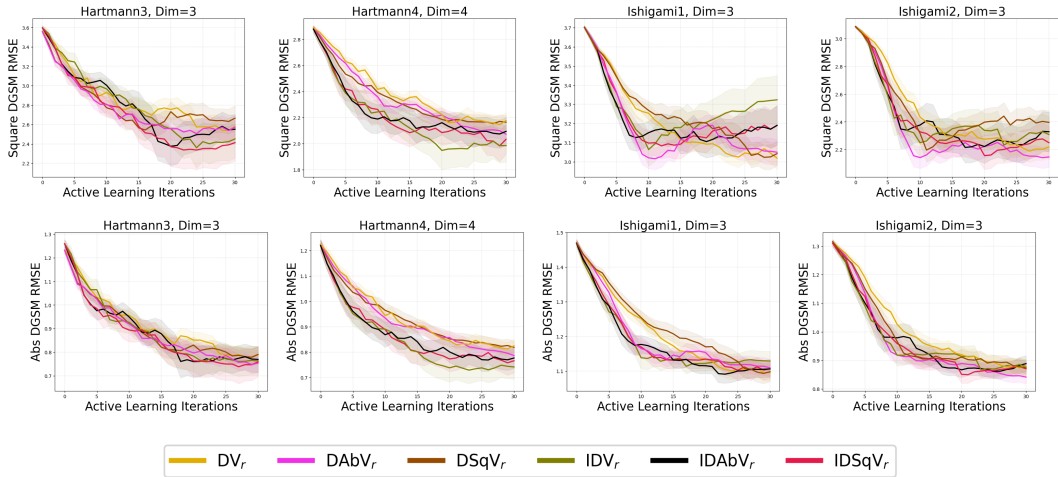

Figure 14: Experiments comparing the global (integrated) variance reduction acquisition function to their local (non-integrated) version. The RMSE of the square DGSMs and absolute DGSMs is reported with the mean and two standard errors of 50 repeats.

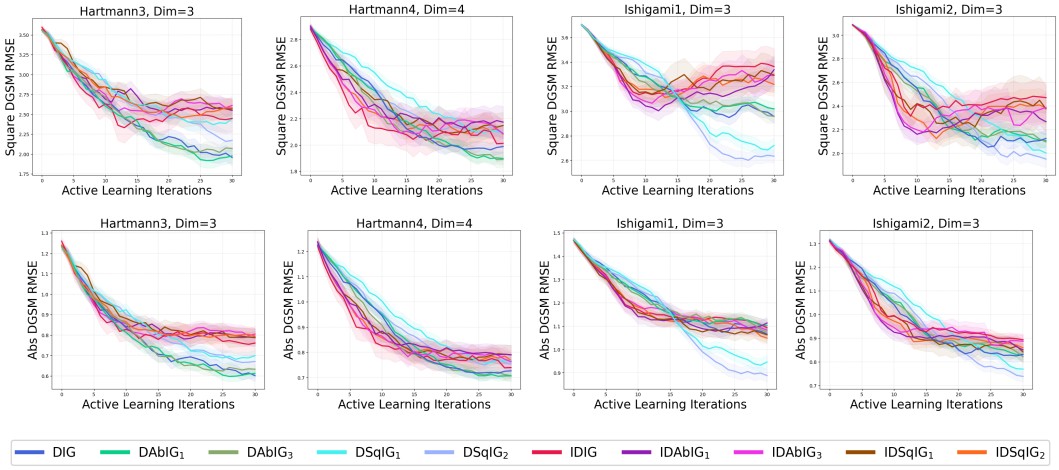

Figure 15: The same results as Fig. 14, for the global (integrated) information gain acquisition functions and their local counterparts.

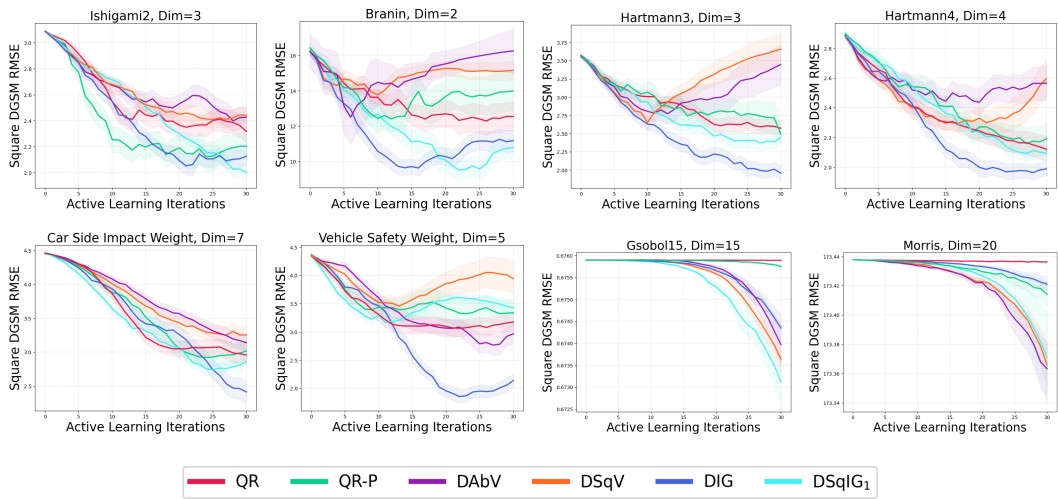

Figure 16: The RMSE of the squared DGSMs is reported as the mean (and two standard errors shaded) over 50 runs of active learning. New results of QR-P baseline are added.

