# OpenReview forum: "Active Learning for Derivative-Based Global Sensitivity Analysis with Gaussian Processes"
_NeurIPS.cc/2024/Conference — NeurIPS 2024 poster_

### Official Review · Reviewer_Uip4 · 2024-07-02

**Soundness:** 3
**Presentation:** 3
**Contribution:** 3
**Rating:** 6
**Confidence:** 4

**Summary:**

This paper considers active learning strategies for global sensitivity analysis of expensive black-box functions to efficiently learn the importance of different input variables. Novel active learning acquisition functions are proposed to target key quantities of derivative-based global sensitivity measures (DGSMs) under Gaussian process surrogate models. The study showcases the application of active learning directly to DGSMs and develops tractable uncertainty reduction and information gain acquisition functions for these measures. Through evaluation on synthetic and real-world problems, it shows that active learning acquisition strategies enhance the sample efficiency of DGSM estimation, especially with limited evaluation budgets.

**Strengths:**

1. The paper is well structured and the main points are clearly outlined.
2. Novel acquisition functions are derived to target key quantities of derivative-based global sensitivity measures (DGSMs) under Gaussian process surrogate models.
3. The performance of the proposed active learning acquisition strategies are evaluated by numerical experimens.

**Weaknesses:**

1. The main goal of global sensitivity analysis is to determine the impact of input variables on the output of a model. The most effective performance metric for this purpose is yet to be identified.
2. There is a need for a practical application that showcases the real-world value of using acquisition functions as presented in the article.

**Questions:**

1. In the right panel of Figure 1, why are the curves in the middle and bottom graphs so similar to each other?
2. In Figure 2, why do some curves rise again as the number of active learning iterations increases?
3. Even if the global sensitivity estimation is very accurate, in practical applications, is it still necessary to use another model to accurately estimate f?

**Limitations:**

Yes

---

> ### Author Rebuttal · Authors · 2024-08-07
>
> Thank you for your review of our paper and for your questions, which we answer below.
>
> **Weaknesses:**
> 1. It is correct that we do use two performance metrics in our paper, RMSEs and NDCGs. As discussed also in the response to reviewer oJrN above, RMSE is the most effective performance metric for this purpose. RMSE is typically more important than NDCG since it shows the ability of the method to converge to the true values of DGSMs. It is also the most commonly used metric in the literature. As far as we are aware, NDCG has not been used before as a performance metric. However, our own experience has been that in some applications, the ranking of the variables is important for practitioners. For instance, parameter selection for downstream tasks. We thus included a ranking metric to provide another perspective into the results. However, ranking metrics sometimes fail to provide insights when several variables are equally important and have the similar DGSMs values. We discussed this case in the appendix and provided results in figure 6. Hence, RMSE generally provides more important insights into the efficacy of various methods.  We will clarify in the paper that RMSE (the typical performance metric in the literature) is the most effective.
>
> 2. Estimating the importance of the variables of black box expensive functions is used in several real-world applications (systems safety [2], biomedicine[3], environmental models [4,5], hydrology [6], and more [1]) where the function evaluations are expensive and the functions are high dimensional, so sample efficiency from active learning is crucial. Prior work has discussed the budget efficiency issue repeatedly [e.g., 7,8,9]. Within our paper, we did apply our methods to 3 real-world problems: the Car Side Impact problem, and two Vehicle Safety problems. These problems use simulations of real safety applications. While the simulations are not costly, we would not be able to rigorously evaluate a large suite of methods as we do in the paper on problems with actual costly evaluations.
>
> **Questions:**
>
> 1. The two figures represent the acquisition function values for the information gain over the gradient and the square of the gradient. The figures show that both acquisition functions score most inputs in the search space similarly, reflecting the fact that in this problem, the best points for learning df/dx also learn (df/dx)**2, and vice versa. This will, of course, not always be the case.
>
> 2. This is a good question that was also asked by reviewer oJrN; see response there for more details. In short, as the method is exploring new regions, it might add points that are not immediately useful, thus temporarily increasing the RMSE by causing an over- or under-fitting of the model. This can temporarily lead to higher RMSE. However, with more data, the model self-corrects and RMSE decreases.It is also important to note that even when this happens, our proposed approaches still outperform the baselines.
>
> 3. To answer your question, we ran experiments to see if our methods that learn the derivatives of f also learn f, or if a separate procedure would be necessary to also estimate f. Conceptually, learning the derivatives of f will also learn f (via integration, by the Fundamental Theorem of Calculus). This is especially true in our setting where we are learning the derivatives from observations of f itself, with a model that is explicitly modeling f alongside its derivatives. However, it is not necessarily the most sample-efficient approach for learning f. The results of the experiments are in Fig. 2 in the rebuttal results, where we evaluate methods on their ability to learn f, by using RMSE of f as the performance metric. Those results show that the derivative information gain (DIG) is actually better at learning f than the f_{VAR} acquisition function that selects points of maximum uncertainty in f. So, the direct answer to your question is no, it is not necessary to use a separate method to accurately estimate f, as estimating the derivatives will in practice also estimate f. The fact that DIG outperforms f_{VAR} can be attributed to DIG being less myopic than f_{VAR}.
>
> [1] Saltelli, Andrea, et al. Global sensitivity analysis: the primer. John Wiley & Sons, 2008.
>
> [2] Qian, Gengjian, et al. "Sensitivity analysis of complex engineering systems: Approaches study and their application to vehicle restraint system crash simulation." Reliability Engineering & System Safety 187 (2019): 110-118.
>
> [3] Wirthl, Barbara, et al. "Global sensitivity analysis based on Gaussian‐process metamodelling for complex biomechanical problems." International journal for numerical methods in biomedical engineering 39.3 (2023): e3675.
>
> [4] Pianosi, Francesca, et al. "Sensitivity analysis of environmental models: A systematic review with practical workflow." Environmental Modelling & Software 79 (2016): 214-232.

---

> > ### Comment · Reviewer_Uip4 · 2024-08-09
> > **Official comment by Reviewer Uip4**
> >
> > Thanks for the response that basically addresses my major concern. I'll raise my score later as a response.

---

> ### Author Response · Authors · 2024-08-12
> **Thank you for your review and feedback on our work**
>
> Dear Reviewer Uip4,
>
> Thank you for your feedback which will help improve the clarity and contribution of the work. We will provide additional context wrt existing acquisition functions, gold-standard evaluation metrics, and the importance of the problem in evaluating expensive-to-compute simulators in the sciences and engineering,
>
> Please let us know if there is anything else you feel that we should touch upon in the CR if accepted.
>
> Finally, as a gentle reminder, we would appreciate if you could raise your score as stated.

---

### Official Review · Reviewer_nwDp · 2024-07-04

**Soundness:** 3
**Presentation:** 4
**Contribution:** 3
**Rating:** 6
**Confidence:** 3

**Summary:**

The authors propose eight acquisition functions for active learning of functions of the gradient of a Gaussian Process, which is motivated by the use of the gradient for global sensitivity analyses. They provide experimental evidence assessing the relative performance of these acquisition functions.

**Strengths:**

The paper’s contribution (a novel application of active learning algorithms and the derivation of several acquisition functions tailored to this application) seems strong. The experiments are thorough, and the paper is well-written and the method clearly presented.

**Weaknesses:**

- Although the application is novel, the proposed acquisition functions are straightforward applications of standard acquisition criteria in the active learning literature.
- The authors provide experimental evidence on the performance of their proposed method(s), but do not provide explicit convergence guarantees or other theoretical results.
- There appear to be many assumptions implicit in the authors’ method, such as that the sensitivity measures are well-approximated by the GP gradient. I would be interested in seeing more discussion of whether these assumptions are required for the effectiveness of their proposed method(s).

**Questions:**

- On pg. 2, the authors state “To learn the squared DGSM, active learning selects points that are adjacent to existing observations, as that adjacency is valuable for the derivative estimate.” This suggests to me that a natural baseline comparison in the experiments would be a heuristic method that selects pairs of points some small distance apart. Do the authors have thoughts on how their proposed acquisition functions would compare to such a heuristic method?
- On line 135, should $m_{\boldsymbol{x}}$ be $m_X$?
- Why is an information gain acquisition function not derived for the absolute gradient?

**Limitations:**

The authors discuss limitations of their work.

---

> ### Author Rebuttal · Authors · 2024-08-07
>
> Thank you for your constructive comments and helpful suggestions. Here are responses to the weaknesses and questions:
>
> **Weaknesses :**
>
> 1. While the variance and IG criteria are known in the literature, formulating them in a computationally efficient and tractable fashion is not straightforward for DGSMs, which is our main technical contribution.
>
> 3. It is correct that the assumption that the GP derivative approximates the gradient of the true function of interest is vital for the success of the methods we develop. This assumption is motivated both practically and theoretically. On the practical side, GPs are indeed the standard surrogate model for sensitivity analysis of blackbox expensive functions; see e.g. [5] from the main text, and citations therein. On the theoretical side, GPs are favored because they are universal approximators, meaning that, with a suitable kernel, they can indeed approximate any arbitrary continuous target function [1]. Thus, requiring the GP gradients to provide a good approximation of the gradients of f is not, in theory, a limiting assumption.
>
> **Questions:**
> 1. Thank you for the suggestion, we implemented and evaluated the suggested idea where we evaluate pairs of points near each other. We generate a quasi-random sequence and then interleave it with points that are small perturbations of the point before it. Specifically, for all n odd, we took x_n from the quasi-random sequence, and then took x_{n+1} = x_n + eps, where eps ~ MVN(0, I*eta) and eta=0.01 (with box bounds scaled to [0, 1]). That is, each dimension is given a small Gaussian perturbation on the value from the previous iteration. Fig. 1 in the rebuttal results shows the performance of this method, which is there called QR-P. In one problem (Ishigami2) it performs very well, though generally the results are comparable to QR without the interleaved pairing. This approach does rely on the perturbation hyperparameter eta, and the “right” choice likely depends on the problem. We will add this ablation study to our paper.
>
> 2. In line 135, we use x to refer to any possible value, so it can be m_X if we are evaluating the mean function on X or can be m_{x*} if we are evaluating it on a new input.
>
> 3. Thank you for the suggestion. We derived and implemented the information gain for the absolute value of the gradient based on an approximation of the entropy of the folded distribution. We show the results in Fig. 3 in the rebuttal results, on the illustrative problem of Fig. 1 from the main text. Please see the response to reviewer oJrN for full details. In short, the approximation uses a truncated Taylor series with an exp() term that is numerically unstable for values of the posterior mean and variance that we run into in practice. Thus, this acquisition function is not a suitable optimization target, but we will add the result and discussion of it to the paper.
>
> [1] CA Micchelli, Y Xu, H Zhang (2006) “Universal Kernels”, Journal of Machine Learning Research 7:2651-2667.

---

> > ### Comment · Reviewer_nwDp · 2024-08-08
> >
> > Thanks to the authors for their response. The authors make the point that although the proposed acquisition functions are known in the literature, formulating them in a computationally efficient and tractable fashion for DGSMs is a contribution. The authors’ rebuttal has also thoroughly answered the questions I raised. For these reasons, I’ve adjusted my rating from 5 to 6. I still feel that the paper could benefit from more extensive discussion of the assumptions implicit in the method.

---

### Official Review · Reviewer_oJrN · 2024-07-07

**Soundness:** 3
**Presentation:** 3
**Contribution:** 2
**Rating:** 6
**Confidence:** 3

**Summary:**

In this paper, the authors study how to select observation data to improve the efficiency of sensitivity analysis. The focus is on measuring sensitivity through a function’s gradient variability. The authors provide a derivation on gradient variance formulae based on Gaussian process surrogates. Various acquisition functions based on gradient, absolute gradient, and squared gradient are described. Empirical evaluations show that the information gain of derivatives and squared derivative perform best in the majority of experiments but in high-dimensional problems, max variance of derivative and squared derivative performs the best. Moreover, the proposed methods can effectively discover the correct ranking for different variables in the sensitivity analysis.

**Strengths:**

1. This work is novel in its systematic study of active learning in DGSM (derivative-based global sensitivity measure). The exposition is clear to me even though I am not an expert in sensitivity analysis.

2. The authors provide comprehensive empirical evaluations of their proposed approaches including both synthetic functions and real-world applications. The results of RMSE estimations show significant improvements over baseline methods.

**Weaknesses:**

1. It seems that some categories are missing some methods. For example, in section 4.3, there is no acquisition function based on information gain for absolute gradient. In the baseline methods, there is no $f_{V_r}$, i.e., maximizing variance reduction on function values. These omissions make the empirical evaluations incomplete.

2. The NDCG (normalized discounted cumulative gain) results are weak compared to the RMSE results. Can the authors elaborate on the practical implication of this? Is RMSE typically more important than NDCG?

**Questions:**

1. What GP kernels did the authors use in their experiments?

2. In Figure 2, RMSE sometimes went up with more active learning iterations. Why is that?

3. Based on the discussion in section 5.1, would it be possible to combine the information gain-based acquisition function with the max variance-based acquisition function to produce a new acquisition function that works well in both low and high-dimensional problems?

**Limitations:**

The authors discussed limitations on some of their acquisition functions.

---

> ### Author Rebuttal · Authors · 2024-08-07
>
> Thank you for your positive feedback and for recognizing the strengths of our paper.  Below we provide answers to your concerns.
>
> **Weaknesses:**
> 1. Thank you for the suggestion to fill out the matrix of acquisition functions. For the baseline f_{V_r}, since our experiments used noiseless function evaluations, the observation at x reduces the GP posterior variance at x to 0; thus f_{V_r} is equivalent to f_{VAR} (max variance of f). We will make sure this is clear in the paper. Information gain for the absolute value of the gradient was indeed missing. We derived and implemented information gain for the absolute derivative based on an approximation of the entropy of the folded normal distribution [1]. We evaluated this acquisition on the illustrative test problem in Figure 1 from the main text; the result is in Fig. 3 in the rebuttal results. While generally, the results look similar to other derivative information gain acquisitions in Fig. 1 of the main text, there is a noticeable discrepancy at x=0.6. This is because of numerical issues. The approximation from [1] uses a truncated Taylor expansion that includes an exponential term that can blow up and become a poor approximation depending on the posterior values. To be useful for acquisition optimization, a new, more accurate and stable approximation of the folded normal distribution entropy will be necessary. We will add this result and discussion of future work to the paper.
>
> 2. RMSE is typically more important than NDCG since it shows the ability of the method to converge to the true values of DGSMs. It is also the most commonly used metric in the literature. As far as we are aware, NDCG has not been used before as a performance metric. However, our own experience has been that in some applications, the ranking of the variables is important for practitioners. For instance, parameter selection for downstream tasks. We thus included a ranking metric to provide another perspective into the results. However, ranking metrics sometimes fail to provide insights when several variables are equally important and have the similar DGSMs values. We discussed this case in the appendix and provided results in figure 6. Hence, RMSE generally provides more important insights into the efficacy of various methods.
>
> **Questions:**
>
> 1. We used an Automatic Relevance Determination (ARD) RBF kernel.
>
> 2. Active learning involves a trade-off between exploring new regions of the input space and exploiting known regions. While the model is still exploring, adding a data point in one part of the space may cause a global adjustment in the model predictions that can cause it to err in another part of the space. This is especially possible early on when the model is still learning the kernel lengthscales, and has the capacity to over- or under-fit the data. With more exploration and data, the model self-corrects and RMSE decreases.
>
> 3. Thank you for the suggestion. This certainly seems possible, and in fact methods that ensemble acquisition functions have recently shown great promise for general-purpose black-box optimization tasks [2]. We will draw connections between those results and this work in our discussion of future work.
>
> [1] M. Tsagris, C. Beneki, H. Hassani (2014) "On the folded normal distribution." Mathematics 2:12-18
>
> [2] R Turner et al. "Bayesian optimization is superior to random search for machine learning hyperparameter tuning: Analysis of the black-box optimization challenge 2020." NeurIPS 2020 Competition and Demonstration Track. PMLR, 2021.

---

> ### Author Response · Authors · 2024-08-13
>
> Dear reviewer,
>
> Thank you again for your review and time.
>
> A major source of concern was that we were missing two baselines. (1) We have clarified that we have in fact already evaluated  $f_{V_r}$ (it is equivalent to $f_\text{VAR}$). (2) We found that IG for the absolute derivative for GPs is not a "baseline" considered by any of the prior literature, and it is in fact a non-trivial acquisition function to derive in our setting.  We have identified an approximation of this quantity based on a Taylor expansion of the entropy of the folded normal, ran experiments, and found that the acquisition function contained discontinuities, rendering it an ill-defined AF for this task.  Thus there is no straightforward "baseline" for this in the literature.
>
> We have also clarified a number of minor points, such as the fact that NDCG is not a common evaluation metric for sensitivity analysis (which is why we did not include it in the main text), and your query about why active learning may decrease RMSE in some situations.

---

### Official Review · Reviewer_uHUu · 2024-07-12

**Soundness:** 4
**Presentation:** 4
**Contribution:** 3
**Rating:** 8
**Confidence:** 4

**Summary:**

The authors develop and compare sequential design criteria for Gaussian-process-based estimation of gradient-based sensitivity metrics to assess the importance of individual variables.

**Strengths:**

The problem of learning variable sensitivities is well-motivated and a matrix of criteria are proposed to address it, from considering different gradient quantities (the gradient itself, its absolute value, and its square) and different measures of sequentially learning them.

This paper clearly addresses an interesting problem and presents itself well.

The numerical experiments are sufficient in rigor and quantity to establish the performance of the proposed method.

**Weaknesses:**

As the authors point out on the checklist, theoretical guarantees on the performance of the acquisition strategies are not available but the numerical experiments are sufficient to establish them.

**Questions:**

I think it's great that the authors considered so many different approaches in their experiments, but it's disappointing that this does not result in practical takeaways for the reader in the conclusion section. If I'm an engineer and want to learn about the sensitivity of my simulator to its parameters, which of your acquisitions would you recommend I start with? Such a discussion would benefit the practioner-reader. Maybe just a sentence or two (beyond what is in 5.1) saying "starting with XYZ is our general purpose recommendation" or some such.

If there is any possibility of adding execution timings to this article I think it would make it more useful to practitioners, regardless of what those timings show.

**Limitations:**

The numerical experiments consider a variety of functions, but consist of very limited budgets.
There appears not to be any discussion of the overhead required to conduct the sequential design.
The authors mention that
"All methods were implemented to be auto-differentiable and, therefore, are efficiently optimized with gradient optimization".
It is great that the methods were implemented in a framework allowing for automatic differentiation.
But this does not guarantee that the optimization will be efficient.
It's true that BO literature often glosses over the overhead required, and that for experiments of very high cost, it is negligible by comparison.
But it is still important to report the overhead execution time.
As they discuss in the checklist, there is some discussion of the complexity order in Appendix B but no reporting of actual execution timings.
This is important to give practitioners thinking about using this method when faced with an expensive-but-not-overwhelmingly-so simulator to decide whether the overhead associated with this method is tolerable relative to their particular application or not.

---

> ### Author Rebuttal · Authors · 2024-08-07
>
> Thank you for your positive feedback and for supporting our paper.  Below we provide answers to your concerns.
>
> **Questions**
>
> 1. We agree that adding a general takeaway would be beneficial for practitioners. We will add the following paragraph to the discussion: “Our general recommendation is to use the information gain of the gradient (DIG) or the information gain of the squared gradient (DSqIG) for low dimensional problems (up to d=10). For high dimensional problems (d>10), we recommend using the variance of the absolute value of the gradient (DAbV) or the variance of the squared gradient (DSqV)”. This will complement the existing discussion in Sec. 5.1 on why the exploratory nature of variance-based approaches are important in high dimensions.
>
> 2. See the table below for the acquisition optimization time for each method across different problems. Generally information gain methods are more expensive, with the squared derivative information gain particularly expensive due to the use of a hypergeometric function in its approximation. However, even the maximum time for that method (214 seconds) is fast relative to the time-consuming function evaluation setting that is the focus of this paper. It is interesting to also note that running time generally increases with dimension but not always. This is because the optimization time will depend on the shape of the acquisition function surface and how hard the optimization is, which is orthogonal to the dimensionality of the problem.
>
>
> | Method | Ishigami1 | Ishigami2 | Hartmann4 | Gsobol6 | Gsobol10 | Gsobol15 | Morris |
> |:--|--|--|--|--|--|--|--|
> | DAbV | 1.37±0.13 | 2.03±0.21 | 0.78±0.05 | 4.26±0.3 | 3.93±0.39 | 5.31±0.63 | 5.87±0.7 |
> | DAbV_r | 2.4±0.3 | 2.06±0.27 | 2.77±0.23 | 9.69±1.19 | 13.31±1.16 | 8.48±1.43 | 3.96±0.42 |
> | DSqV_r | 2.84±0.64 | 2.3±0.37 | 2.0±0.21 | 6.66±0.88 | 8.8±1.77 | 6.29±0.9 | 9.0±2.29 |
> | DV_r | 1.43±0.2 | 1.32±0.23 | 1.66±0.14 | 3.43±0.49 | 10.56±1.78 | 6.32±1.06 | 3.54±0.51 |
> | DSqV | 1.58±0.22 | 2.44±0.25 | 0.81±0.07 | 4.23±0.62 | 12.18±4.23 | 16.1±2.03 | 6.95±1.0 |
> | DIG | 8.59±1.15 | 9.45±0.99 | 8.09±1.09 | 16.75±0.46 | 24.6±2.5 | 28.15±4.75 | 13.92±3.78 |
> | fIG | 0.43±0.03 | 0.44±0.03 | 0.41±0.03 | 0.37±0.01 | 0.29±0.04 | 0.17±0.02 | 0.16±0.01 |
> | DV | 0.62±0.08 | 0.79±0.09 | 0.1±0.01 | 0.31±0.01 | 0.45±0.07 | 0.34±0.04 | 0.16±0.01 |
> | fVAR | 0.44±0.03 | 0.48±0.03 | 0.37±0.02 | 0.39±0.02 | 0.19±0.01 | 0.21±0.02 | 0.2±0.02 |
> | DSqIG | 96.3±19.52 | 100.95±21.65 | 151.56±37.84 | 214.39±30.84 | 180.5±24.79 | 122.27±18.61 | 95.24±28.67 |

---

> > ### Comment · Reviewer_uHUu · 2024-08-08
> >
> > Thanks for these updates; I'm going to increase my score in response.

---

### Author Rebuttal · Authors · 2024-08-07

We thank all of the reviewers for their constructive reviews. We have included new results in the rebuttal to address the major questions from each review:

1. Running times (Reviewer uHUu): The table in that review response provides running times for all of the active learning methods, for 7 of the benchmark problems.

2. Information gain for the absolute DGSM (Reviewers oJrN and nwDp): We added an implementation of this method, and Fig. 3 in the rebuttal results shows how it works in the test problem of Fig. 1 from the main text. In the detailed response below, we explain why this approach has numerical issues that make it a poor optimization target.

3. A new baseline method using paired sampling (Reviewer nwDp): We implemented the new baseline method suggested by the reviewer (here called QR-P; details below), and give the results for 8 of the benchmark problems in Fig. 1 in the rebuttal results.

4. Does learning DGSMs also learn f (Reviewer Uip4): We evaluated the active learning methods with RMSE of f, instead of RMSE of the DGSMs, and found that derivative information gain does also learn f, better than the max-variance-of-f method (Fig. 2 in the rebuttal results). So subsequent evaluation to learn f in addition to the DGSMs is not necessary.

These additional analyses have strengthened the paper, for which we again thank the reviewers.

---

### Decision · Program_Chairs · 2024-09-25

**Decision:**

Accept (poster)

**Comment:**

The paper introduces novel active learning acquisition functions for global sensitivity analysis of expensive black-box functions, focusing on derivative-based global sensitivity measures (DGSMs) using Gaussian process models. The proposed methods significantly improve sample efficiency and accuracy in sensitivity analysis, particularly with limited evaluation budgets, as demonstrated through comprehensive synthetic and real-world evaluations. The rebuttals have resolved other reviewer concerns, and the proposed changes are straightforward for the camera ready version. I concur with the reviewers' consensus on acceptance.